# Large-scale DNA demethylation occurs in proliferating ovarian granulosa cells during mouse follicular development

Tomoko Kawai[1], JoAnne S. Richards[2] & Masayuki Shimada [1✉]

During ovarian follicular development, granulosa cells proliferate and progressively differentiate to support oocyte maturation and ovulation. To determine the underlying links between proliferation and differentiation in granulosa cells, we determined changes in 1) the expression of genes regulating DNA methylation and 2) DNA methylation patterns, histone acetylation levels and genomic DNA structure. In response to equine chorionic gonadotropin (eCG), granulosa cell proliferation increased, DNA methyltransferase (DNMT1) significantly decreased and Tet methylcytosine dioxygenase 2 (TET2) significantly increased in S-phase granulosa cells. Comprehensive MeDIP-seq analyses documented that eCG treatment decreased methylation of promoter regions in approximately 40% of the genes in granulosa cells. The expression of specific demethylated genes was significantly increased in association with specific histone modifications and changes in DNA structure. These epigenetic processes were suppressed by a cell cycle inhibitor. Based on these results, we propose that the timing of sequential epigenetic events is essential for progressive, stepwise changes in granulosa cell differentiation.

[1] Laboratory of Reproductive Biology, Graduate School of Integrated Sciences for Life, Hiroshima University, Higashi-Hiroshima, Japan. [2] Department of Molecular and Cellular Biology, Baylor College of Medicine, Houston, Texas, USA. ✉email: mashimad@hiroshima-u.ac.jp

Several epigenetic mechanisms have been identified in cells as they differentiate, including stem cell and embryonic specification, epithelial to mesenchymal changes in cancer cells, and immune cell lineages[1–3]. Specific factors that are known to control epigenetic processes include DNA methyltransferases (DNMTs), which act to methylate specific regions of the genome, especially CpG-rich promoter regions, and hence block and impair transcription[4,5]. Once the DNA methylation patterns in a specific cell type are established, they are copied to a daughter cell by the functions of the DNMT1/PCNA/UHRF1 complexes[6]. Methylation events are reversed and regulated by TET enzymes (Tet methylcytosine dioxygenases) that convert DNA methylcytosine to 5-hydroxymethylcytosine[7,8]. Collectively, these pathways control or modify the methylated CpG islands found frequently in promoters of key regulatory genes to induce specific gene expression patterns.

The development of ovarian follicles capable of ovulating and releasing a fertilizable oocyte is a complex process that is regulated by intrafollicular factors as well as peripheral hormones, including follicle stimulating hormone (FSH), which binds to the FSH receptor and regulates granulosa cell proliferation and functions that are essential supporters/regulators of oocyte growth and maturation[9–11]. Specifically, during the transition of secondary follicles (SFs) to antral follicles, granulosa cells express constant levels of the FSH receptor and produce estrogen but little to no progesterone, a marker of differentiated granulosa cells that have terminally differentiated into luteal cells[9]. During the growth of antral follicles to the preovulatory stage, proliferating granulosa cells differentiate in response to FSH and acquire the ability to respond to the ovulatory surge of luteinizing hormone (LH)[9]. The FSH-mediated differentiation process includes the induction of ~500 genes[12,13], including Lhcgr, which encodes the LH receptor that is essential for LH-mediated ovulation[14,15].

FSH-stimulated granulosa cell proliferation is dependent on the expression of the cell cycle regulator cyclin D2[16,17]. Mutant mice lacking the Ccnd2 gene that encodes cyclin D2 exhibit female infertile phenotypes characterized by small follicles, depletion of granulosa cells, and altered gene expression patterns, including the lack of Lhcgr[18], suggesting that cell proliferation might be involved in the differentiation of granulosa cells and the large-scale epigenetic changes that occur during follicular development and ovulation. Specifically, DNA methylation of the promoter region of Lhcgr in granulosa cells is ~50% in small follicles; however, methylation decreases to <20% in proliferating granulosa cells of preovulatory follicles (PFs)[19]. In contrast, Lhcgr is constitutively expressed in theca cells of growing and PFs, and the DNA methylation rate is low[19], indicating that the DNA methylation pattern might be altered in granulosa cells but not in theca cells during the marked transition of granulosa functions in PFs. Canon et al.[20] reported that proliferating B cells have much lower rates of DNA methylation in marker genes of plasma cells than nonproliferating B cells. However, the relationship between cell proliferation and epigenetic regulation of cell differentiation remains unclear.

The reduction of DNA methylation in the Lhcgr promoter region of granulosa cells is dependent on not only FSH but also the de novo synthesis of retinoic acid (RA) and SMAD pathways[19]. RA and SMADs are known factors involved in cell fate determination due to their impact on epigenetic modifications[21,22]. Therefore, the studies described herein were undertaken to analyze granulosa cells as a model to determine the underlying mechanisms by which DNA methylation changes dramatically in highly proliferative cells.

## Results

### Temporal changes occur in the expression of DNMT1 and other epigenetic regulatory factors in proliferating granulosa cells at specific stages of follicular development. The induction of follicle development by equine chorionic gonadotropin (eCG)

was used as a general treatment for superovulation in mice. The experimental design in detail was mentioned in Supplemental Notes 1–6. In granulosa cells of PFs isolated from mouse ovaries 48 h after eCG injection, the expression levels of Dnmt1 were significantly decreased compared with those in antral follicles isolated from mice before eCG injection (Fig. 1a; 0 h). In contrast, the expression of Dnmt3l was significantly decreased in granulosa cells in antral follicles isolated from mouse ovaries before eCG injection compared with that in SFs. Tet2 expression was significantly induced in antral follicles (eCG 0 h) from SFs; a further significant induction was observed in PFs at 48 h after eCG injection (Fig. 1a). The expression levels of Dnmt3a, Uhrf1, Tet1, and Tet3 did not change significantly during PF growth (Fig. 1a). Changes in DNMT1 and Tet methylcytosine dioxygenase 2 (TET2) in granulosa cells during follicular development were also analyzed by western blotting (Fig. 1b, c). DNMT1 significantly decreased in granulosa cells at 24 and 48 h after eCG injection compared with that before eCG injection (eCG 0 h). The level of TET2 was dramatically higher in granulosa cells at 48 h after eCG injection than before eCG injection (eCG 0 h).

To observe the cell-specific localization of DNMT1, TET2, and PCNA (a marker of proliferation) in granulosa cells during the S-phase of the mitotic cell cycle, immunofluorescence analyses were performed. Most of the cells surrounding oocytes (granulosa cells in SFs and cumulus cells in antral follicles and PFs) were DNMT1 positive (Fig. 1d and Supplementary Fig. 1). A high level of DNMT1 was also detected in granulosa cells of antral follicles, whereas DNMT1 was significantly decreased in granulosa cells of PFs (48 h after eCG injection). In contrast, the number of PCNA-positive cells was significantly increased in both granulosa cells and cumulus cells in response to eCG treatment (Fig. 1d and Supplementary Fig. 1). Most PCNA-positive cells were also DNMT1 positive in cumulus cells and in granulosa cells of SFs. However, the number of DNMT1-positive cells relative to PCNA-positive cells significantly decreased in a stage-dependent manner in granulosa cells as they differentiated (Fig. 1d and Supplementary Fig. 1). TET2-positive cells were localized in follicles from the secondary to preovulatory stages. However, TET2-positive cells were significantly increased in both cumulus cells and granulosa cells at 48 h after eCG injection (Fig. 1d and Supplementary Fig. 1). TET2 expression was observed in both proliferating (PCNA+) and nonproliferating (PCNA−) cells.

### The induction of DNMT1 is significantly decreased in S-phase granulosa cells during follicular development. To determine the relationship between cell cycle progression and the expression level of DNMT1 in more detail, the stage of the cell cycle was analyzed by propidium iodide (PI) staining, and the expression level of DNMT1 was analyzed by flow cytometry in granulosa cells collected at specific time points. Dot blot graphs were made on the basis of FSC-A (forward scatter area) and SSC-A (side scatter area), and 20,000 granulosa cells were gated (Supplementary Fig. 2a, b). After eCG injection, the proliferative activity increased in granulosa cells; the percentage of cells in the G0/G1 phase was significantly decreased, and the percentages of cells in S-phase and G2/M-phase were significantly increased (Fig. 2a). In each cell cycle stage, the fluorescence intensity of DNMT1 was detected by flow cytometric analyses (Fig. 2b). DNMT1 increased from the G0/G1 stage to the S-phase. However, the fluorescence intensity of DNMT1 was significantly decreased in S-phase granulosa cells collected from large antral follicles (PFs) at 48 h after eCG injection (Fig. 2b). The lower threshold of the DNMT1 level at S-phase was set as 5% fewer expressed cells at eCG 0 h, as a statistically significant change (Fig. 2c, red dashed line). The number of cells expressing low levels of DNMT1 (red collar) at

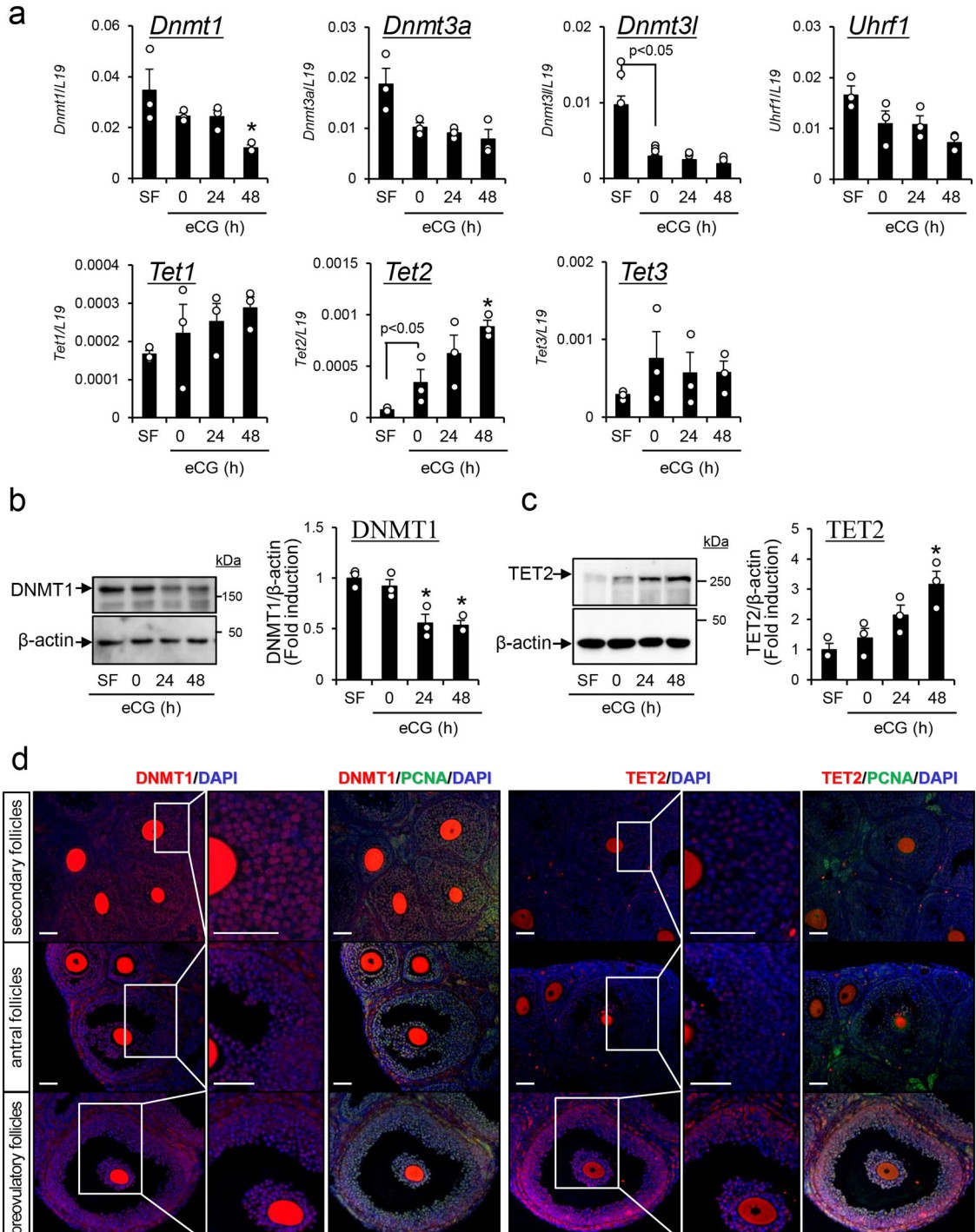

**Fig. 1 Kinetic changes in the expression levels of genes involved in epigenetic regulatory mechanisms in mouse granulosa cells during follicular development. a** The expression levels of epigenetic-related genes in granulosa cells. Levels of mRNA were normalized to that of *L19*. Data represents the mean ± SEM ($n = 3$ biological replicates). Significant differences were observed between SF and eCG at 0 h ($p < 0.05$). *Significant differences were induced by eCG injection compared with before eCG injection (0 h) ($p < 0.05$). SF, secondary follicles. **b**, **c** Temporal changes of DNMT1 (**b**) and TET2 (**c**) in granulosa cells of secondary follicles (SF) with multilayered granulosa cells and antral follicles before or after eCG stimulation. β-Actin was used as a loading control. The results are representative of three independent experiments. Immunoblots shown are representative of three independent experiments. In each experiment, the different protein samples were used ($n = 3$ biological replicates). Quantitative expression of DNMT1 and TET2 relative to the expression of β-actin (control), as determined by western blotting. The SF value was set as 1 and the data are expressed as fold induction. The values are the mean ± SEM ($n = 3$ biological replicates). Significant differences were observed between SF and eCG at 0 h ($p < 0.05$). *Significant differences were induced by eCG injection compared with before eCG injection (0 h) ($p < 0.05$). Full-length blotting images of DNMT1, TET2, and β-actin are shown in Supplementary Fig. 8. **d** The localization of DNMT1, TET2, and PCNA in secondary follicles (SFs) and antral follicles (AFs) of 3-week-old immature mice or preovulatory follicles (PFs) of ovaries collected 48 h after eCG injection. DNMT1 and TET2 were visualized with Cy3, PCNA was visualized with FITC, and nuclei were stained with DAPI in the ovaries of 3-week-old mice treated with or without eCG. All of different length scale bars show 50 μm.

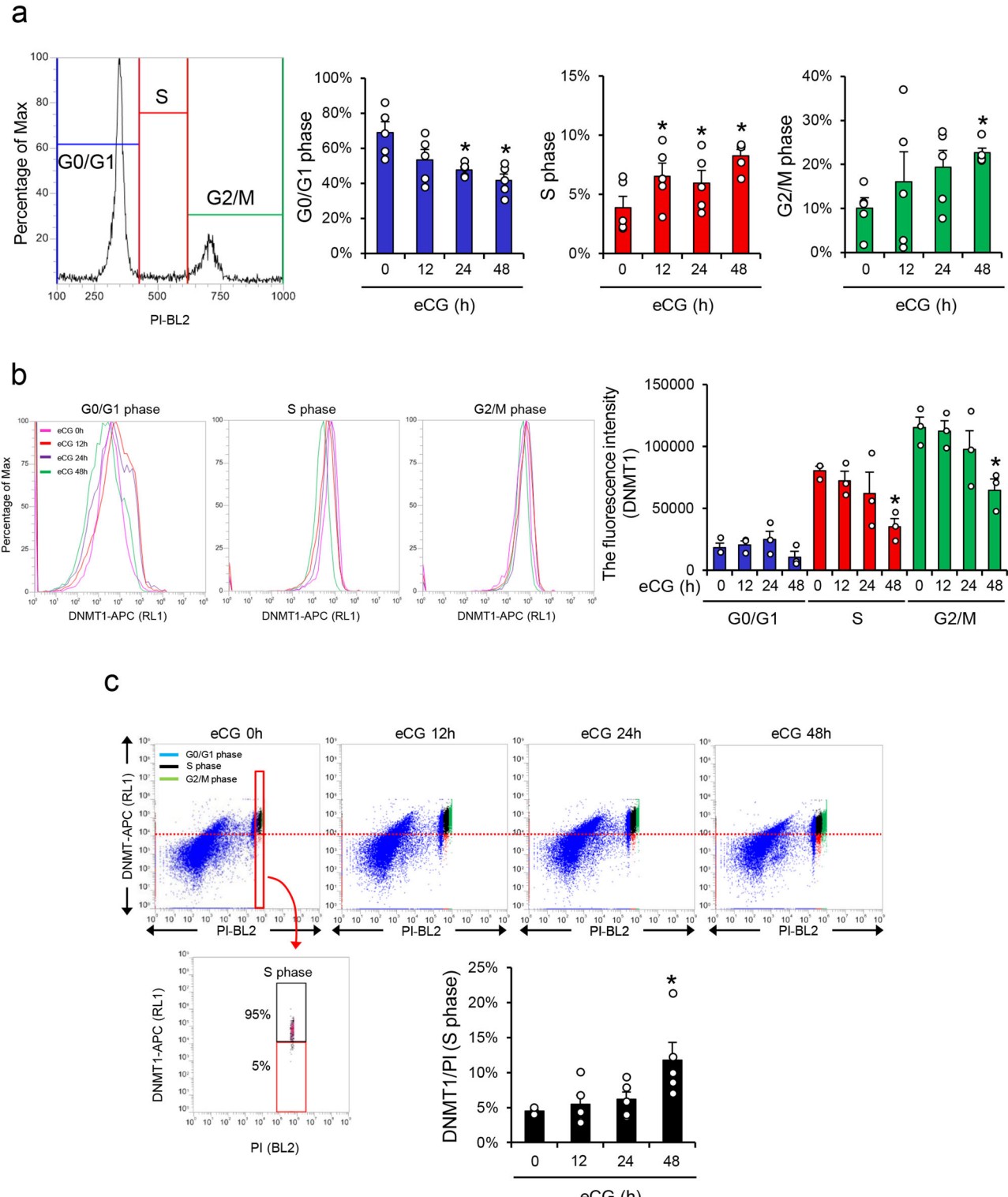

S-phase was significantly increased at 48 h after eCG injection (Fig. 2c).

**Genome-wide DNA demethylation occurred in the promoter regions of genes in granulosa cells of PFs**. The comprehensive changes in the DNA methylation status in granulosa cells during follicular development were analyzed by methylated DNA immunoprecipitation coupled with next-generation sequencing (MeDIP-Seq) analyses. The absolute methylation signals (AMS)

were used for the analysis of the percentage of DNA methylation of the promoter region in each gene[23,24]. The significantly demethylated genes in granulosa cells of PFs (48 h after eCG injection) compared with those in SFs represented 13.6% of the whole genome. The most dramatic changes in methylation status occurred in promoter regions (37.87%) and satellite regions (29.73%) of the genome (Fig. 3a). In contrast, increases in methylated regions in the whole genome of granulosa cells during follicular development were minor (Fig. 3b). The genes in which

**Fig. 2 Relationship between the level of DNMT1 and cell cycle stages in granulosa cells during the follicular development process. a**, **b** Positive fluorescence signal of PI (**a**) or DNMT1 (**b**) in granulosa cells collected from ovaries before or after eCG injection. Granulosa cells were collected from 3-week-old mice and 3-week-old mice at 12, 24, or 48 h after eCG injection. The granulosa cells were incubated with Cy3-tagged anti-DNMT1 antibody and PI (propidium iodide), and then the fluorescence signals were detected by flow cytometry analysis. **a** Each percentage of G0/G1 phase cells, S-phase cells, or G2/M-phase cells is shown in the bar graph. **b** The mean fluorescence intensity was analyzed in DNMT1-positive cells. The unstained sample was used to set a region for the cells that were not stained (negative). All of experiments are represented as means ± SEM, $n = 3$ (Fig. 2b) or $n = 5$ (Fig. 2a) biological replicates. *Significant differences were observed between eCG at 0 h and eCG-primed mice ($p < 0.05$). Significant differences in percentage values were transformed into normally distributed numbers by angle transformation and then analyzed by one-way ANOVA. Tukey–Kramer was used as post hoc test. **c** Variations in positive fluorescence signals of DNMT1 in granulosa cells in each cell cycle stage. The dot plot graph of double-positive granulosa cells with DNMT1 and PI. The percentage of cells that expressed lower levels of DNMT1 at S-phase is shown in the bar graph. The lower threshold of DNMT1 level at S-phase was set as 5% of lower expressed cells at eCG 0 h (red line). The cells that expressed lower levels of DNMT1 are shown in the red box and cells with higher DNMT1 expression are shown in the black box at S-phase. Experiment is represented as means ± SEM, $n = 5$ biological replicates. *Significant differences were observed between eCG at 0 h and eCG-primed mice ($p < 0.05$). Significant differences in percentage values were transformed into normally distributed numbers by angle transformation and then analyzed by one-way ANOVA. Tukey–Kramer was used as post hoc test.

their promoter regions were demethylated accounted for more than 30% of all chromosomes, including the X chromosome (Fig. 3c). Cluster analysis also showed that as an overall trend, promoter regions were demethylated in a stage-dependent manner in granulosa cells during follicular development (Fig. 3d). The changes in the methylation status of specific genes highly expressed in granulosa cells during follicular development (Fig. 3e) and during the ovulation process (Fig. 3f) are listed. The methylation status did not change dramatically in the genes *Ccnd2*, *Foxo3*, *Fshr*, and *Ar*, which are highly expressed in SFs and maintained in PFs[25,26]. However, the methylation levels dramatically decreased to approximately half that in the *Amhr2*, *Gja1*, *Nr5a1*, and *Lhcgr* genes, in which transcription is known to be increased after eCG injection[25,27,28]. The *Areg*, *Ereg*, *Btc*, *Star*, *Cyp11a1*, *Hsd3b*, *Ptx3*, and *Ptgs2* genes, which regulate ovulation and were induced after hCG injection[29,30], were also dramatically demethylated in granulosa cells of PFs. Annotation analysis of demethylated genes was performed to identify the predicted functions in biological processes and the results are shown in Supplementary Tables 1 and 2.

**Validation of epigenetic modifications (DNA methylation status, H3K27 acetylation, and opened chromatin structure) in candidate genes identified by MeDIP-seq analysis in granulosa cells.** From the top nine most highly demethylated genes listed in Supplementary Table 3, *Stk36* and *Trnau1ap*, as well as *Lhcgr*, were selected for validation of their promoter methylation status by bisulfite sequence analyses. In the promoter regions of the *Stk36*, *Trnau1ap*, and *Lhcgr* genes, the percentage of methylated CpG sites was significantly decreased in a time-dependent manner after eCG injection (Fig. 4a). Promoter demethylation of these genes was associated with significantly increased PCR products by formaldehyde-assisted isolation of regulatory elements-quantitative PCR (FAIRE-qPCR) (opened chromatin status of promoter regions, OCRs) (Fig. 4b). The acetylation of K27 in histone H3 (H3K27ac) was also increased in the promoter regions of the *Stk36*, *Trnau1ap*, and *Lhcgr* genes (Fig. 4c). Furthermore, the expression of mRNA for each gene, especially *Lhcgr*, was also significantly increased in granulosa cells after eCG treatment (Fig. 4d). Changes in DNA structure, histone acetylation and transcription were also significantly induced in the other seven demethylated genes listed in Supplementary Table 3 (Supplementary Fig. 3a–c).

**Histone modification and enhanced gene expression occurred only after hCG injection following eCG-induced demethylation and altered chromatin structure in LH target genes.** The promoter regions of LH target genes (*Areg*, *Ereg*, *Btc*, *Star*, *Cyp11a1*,

and *Ptgs2*) were demethylated by eCG stimulation in granulosa cells (Fig. 3f). The chromatin structure examined by FAIRE-qPCR assays was also opened in the promoter regions of these genes in granulosa cells 48 h after eCG injection (just before hCG injection) (Supplementary Fig. 4a). However, the levels of H3K27 acetylation of these LH target genes in granulosa cells were not significantly changed by eCG injection (Supplementary Fig. 4b). The injection of hCG following 48 h after eCG stimulation was required to increase the levels of H3K27 acetylation in the promoter regions of LH target genes (Supplementary Fig. 4b). The higher total level of H3K27 acetylated histone H3 was maintained by 4 h after hCG injection (Supplementary Fig. 5). Increased acetylation occurred simultaneously with significantly increased expression of the LH target genes in eCG-primed granulosa cells exposed to an ovulatory dose of hCG (Supplementary Fig. 4c).

**A cell cycle inhibitor significantly suppressed DNA demethylation in FSH- and testosterone-stimulated granulosa cells.** To determine the mechanisms that mediate promoter demethylation of the *Stk36*, *Trnau1ap*, and *Lhcgr* genes in granulosa cells, the percentage of CpG islands in which all CpG sites were not methylated in each promoter region was calculated based on bisulfite sequence analysis. The percentage of nonmethylated CpG islands of *Stk36*, *Trnau1ap*, and *Lhcgr* significantly increased in granulosa cells obtained from SFs to PFs in vivo (Fig. 5a). When granulosa cells collected from immature mice were cultured with FSH and testosterone, which are known as inducers of granulosa cell differentiation in vitro[19], an increase in nonmethylated CpG islands was also observed (Fig. 5b). The ratio of demethylated CpG islands was significantly reduced by the addition of aphidicolin, a cell division inhibitor (Fig. 5b). Likewise, the expression of *Stk36*, *Trnau1ap*, and *Lhcgr* was increased in granulosa cells cultured with FSH and testosterone (FSH + T), whereas the induction of these genes was significantly suppressed by aphidicolin in a dose-dependent manner (Fig. 5c). These results indicated that demethylation was linked to cell cycle progression and was possibly related to changes in the cellular levels of DNMT1 and TET2 (Fig. 1).

**Oocyte-secreted factors blocked the reduction of *Dnmt1* expression and the induction of *Tet2*, whereas RA overcame the negative effects.** To determine factors that impact epigenetic events mediated by DNMT1 and TET2 during granulosa cell differentiation, granulosa cells were cultured with FSH and testosterone (FSH + T) in the presence of serum. The expression of *Dnmt1* mRNA was significantly decreased, whereas the expression of *Tet2* mRNA was significantly increased (Fig. 6a, b). However, when granulosa cells were cocultured with denuded

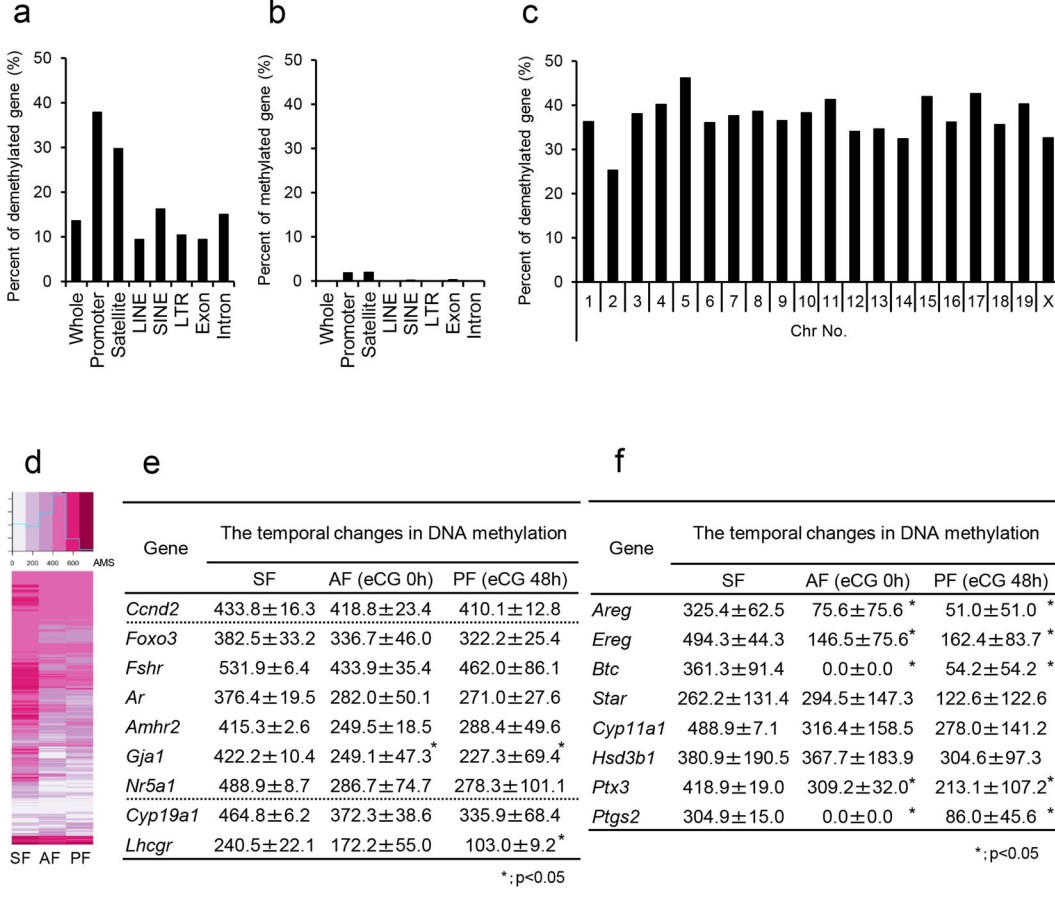

**Fig. 3 Comprehensive analysis of genome-wide DNA methylation status in granulosa cells during follicular development. a, b** The percentage of genes that were significantly demethylated (**a**) or methylated (**b**) during follicular development in each genomic region. **c** The percentage of significantly demethylated genes in promoter regions of each chromosome. **d** Clustering analysis of demethylated genes in promoter regions of granulosa cells during follicular development. SF, granulosa cells in secondary follicles with multilayered granulosa cells; AF, granulosa cells in antral follicles (3-week mice not treated with eCG); PF, granulosa cells in preovulatory follicles (at 48 h after eCG injection). **e, f** Temporal changes in DNA methylation of marker genes (**e**) and LH-targeting genes (**f**) in granulosa cells during the follicle development process. The values are represented as the mean ± SEM ($n = 3$ biological replicates). The mean ± SEM values from MeDIP-seq analysis are absolute methylation signals (AMS) in specific promoter regions containing CpG sites. For genes with multiple specific promoter regions, including CpG sites, the average value was calculated. The significant differences between values in antral follicles (3-week mice not treated with eCG) or preovulatory follicles (at 48 h after eCG injection) and that in granulosa cells of secondary follicles (SF) were analyzed ($p < 0.05$).

oocytes, the effects of FSH + T were dramatically reduced (Fig. 6a). The reduced expression of *Dnmt1* and the induction of *Tet2* were also regulated by additional treatment with GDF9 (Fig. 6b). However, the addition of RA to the medium overcame the negative effects mediated by coculture with denuded oocytes or treatment with GDF9 (Fig. 6a, b).

**RA is a key regulator of DNMT1 expression in S-phase granulosa cells**. To determine how the expression levels of *Ccnd2*, *Dnmt1*, *Tet2*, and *Lhcgr* were regulated in granulosa cells during follicular development, granulosa cells were cultured with FSH + testosterone (FSH + T) and/or RA in the absence of serum (which contains retinol and an RA precursor). Treatment with RA alone significantly decreased *Dnmt1* expression and increased *Tet2* expression (Fig. 7a). The DNA methylation ratios in CpG islands in the promoter regions of *Stk36* and *Trnau1ap* were slightly but not significantly decreased; however, further significant reduction was induced by the addition of FSH + T to the RA-containing medium (Supplementary Fig. 6). On the other hand, FSH + T alone (without RA) did not induce the suppression of *Dnmt1* or demethylation of CpG islands in the promoter

regions of *Stk36* and *Trnau1ap* when granulosa cells were cultured without fetal beef serum. The expression patterns of *Stk36* and *Trnau1ap* were similar to those in *Lhcgr* (Fig. 7a). The percentage of cells that expressed low levels of DNMT1 at S-phase was significantly increased by the addition of RA to FSH + T-containing serum-free medium. Using serum-containing medium, FSH + T significantly increased the percentage of cells that expressed low levels of DNMT1. The induction was significantly suppressed by the additional 4MP (ADH inhibitor), which reduced the cellular levels of RA (Fig. 7b, c). Collectively, these results indicate that RA exerts potent regulatory effects on the expression of DNMT1 in granulosa cells.

## Discussion
Development-related changes in tissue specification depend on processes that tightly regulate cell proliferation and differentiation. Although transcription factors play a key role in gene activation, gene expression patterns in each cell also depend on changes in cell-specific chromatin structure, notably the open chromatin structure in the S-phase of the cell cycle and in promoter and enhancer regions of activated genes[31–33]. Genomic

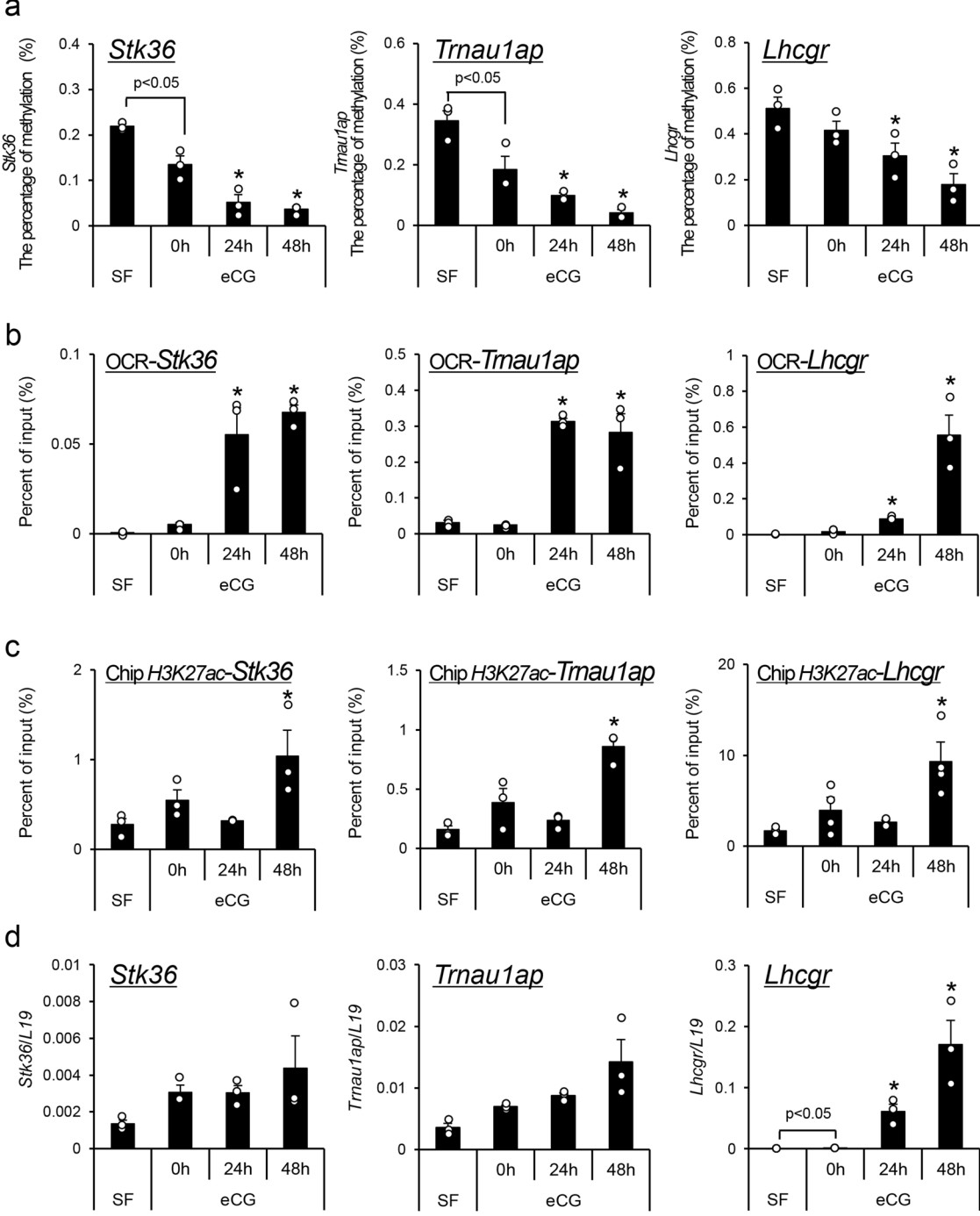

**Fig. 4 Changes in epigenetic regulation of the demethylated genes *Stk36*, *Trnau1ap*, and *Lhcgr* in granulosa cells during follicular development. a** The temporal changes in methylation of promoter regions were analyzed by bisulfite sequencing. Granulosa cells were collected from secondary follicles with multilayered granulosa cells or antral follicles (3-week mice not treated with eCG) before or after eCG injection. Values are represented as the mean ± SEM ($n = 3$ biological replicates). Significant differences were observed between SF and eCG at 0 h ($p < 0.05$). *Significant differences were induced by eCG injection compared with before eCG injection (0 h) ($p < 0.05$). SF, secondary follicles. **b**, **c** The kinetic changes of the opened chromatin region (OCR) detected by FAIRE-qPCR (**b**) or the acetylated H3K27 detected by ChIP assay (**c**) in the promoter region of each gene in granulosa cells of secondary follicles with multilayered granulosa cells or antral follicles (3-week mice not treated with eCG) before or after eCG injection. The SF value was set as 1 and the data are expressed as fold induction. Values are represented as the mean ± SEM ($n = 3$ biological replicates). Significant differences in percentage values were transformed into normally distributed numbers by angle transformation and then analyzed by one-way ANOVA. Tukey–Kramer was used as post hoc test. *Significant differences were induced by eCG injection compared with before eCG injection (0 h) ($p < 0.05$). **d** Kinetic changes of the expression level of each gene in granulosa cells. Levels of mRNA were normalized to that of *L19*. Values are represented as the mean ± SEM ($n = 3$ biological replicates). Significant differences were observed between SF and eCG at 0 h ($p < 0.05$). *Significant differences were induced by eCG injection compared with before eCG injection (0 h) ($p < 0.05$).

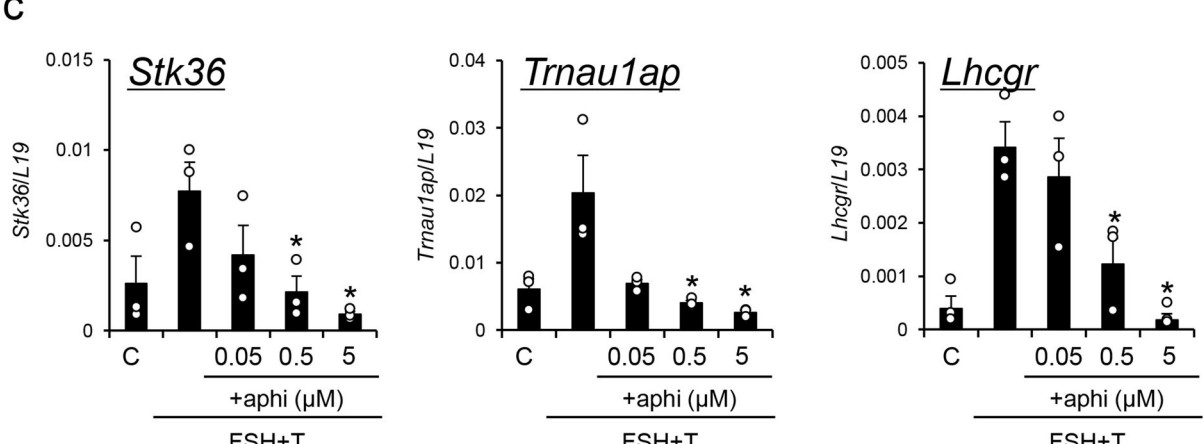

| | SF | eCG 0h | eCG 24h | eCG 48h |
|---|---|---|---|---|
| *Stk36* (%) | 58.3±4.2 | 70.8±4.2 | 79.2±4.2* | 87.5±0.0* |
| *Trnau1ap* (%) | 37.5±7.2 | 45.8±4.2 | 66.7±8.3* | 75.0±0.0* |
| *Lhcgr* (%) | 8.3±7.2 | 20.8±4.2 | 37.5±7.2* | 45.8±4.2* |

*; $p < 0.05$

| | C | FSH+T | FSH+T+aphi |
|---|---|---|---|
| *Stk36* (%) | 20.8±11.0 | 62.5±7.2 | 25.0±12.5* |
| *Trnau1ap* (%) | 45.8±11.0 | 62.5±7.2 | 41.7±11.0 |
| *Lhcgr* (%) | 9.4±6.0 | 28.1±3.1 | 6.3±3.6* |

*; $p < 0.05$

structural variations are regulated by the differences in DNA methylation and histone modifications[34] that are controlled by hormones and signaling pathways that impact cell proliferation and differentiation[20,35]. In this study, we document that hormone-regulated proliferation and differentiation of granulosa cells during follicular development from the secondary to preovulatory stage are dependent on specific temporal changes in DNA methylation and histone acetylation (Fig. 8).

In proliferating cells, DNA replication points are formed at G1 phase, in which genomic AT-rich regions are open[35]. At S-phase, proteins involved in DNA replication, such as DNA helicase and PCNA, are recruited to these regions[36]. DNMT1 is also observed in

**Fig. 5 The effect of cell cycle inhibitor on the demethylation of *Stk36*, *Trnau1ap*, and *Lhcgr*, and on their expression in granulosa cells during follicular development. a** The percentage of nonmethylated CpG islands in the promoter regions of *Stk36*, *Trnau1ap*, and *Lhcgr* in granulosa cells during the follicular development process. ●, Methylated cytosine. ○, Unmethylated cytosine. SF, secondary follicles. Values are represented as the mean ± SEM ($n = 3$ biological replicates). *There were significant differences compared to SF ($p < 0.05$). Significant differences in percentage values were transformed into normally distributed numbers by angle transformation and then analyzed by one-way ANOVA. Tukey–Kramer was used as post hoc test. **b** The effects of the cell cycle inhibitor aphidicolin (aphi) on the appearance of nonmethylated CpG islands in the promoter regions of *Stk36*, *Trnau1ap*, and *Lhcgr* in granulosa cells cultured with FSH and testosterone (FSH + T) in the presence of 1% FBS. ●, Methylated cytosine. ○, Unmethylated cytosine. C, without any hormones (control). Values are represented as the mean ± SEM ($n = 3$ biological replicates). *Significant differences were observed between the control group and the aphidicolin treatment group in the presence of FSH + testosterone and 1% FBS ($p < 0.05$). Significant differences in percentage values were transformed into normally distributed numbers by angle transformation and then analyzed by one-way ANOVA. Tukey–Kramer was used as post hoc test. **c** The effects of aphidicolin on the expression of *Stk36*, *Trnau1ap*, and *Lhcgr* in granulosa cells. Granulosa cells were cultured without any hormones (control C) or with FSH (100 ng/ml) and testosterone (T; 10 ng/ml) without or with different concentrations of aphidicolin (aphi: 0.05, 0.5, 5 μM) in the presence of 1% FBS. Levels of mRNA were normalized to that of *L19*. Values are represented as the mean ± SEM ($n = 3$ biological replicates). *Significant differences were observed between the control group and the aphidicolin treatment group in the presence of FSH + testosterone and 1% FBS ($p < 0.05$).

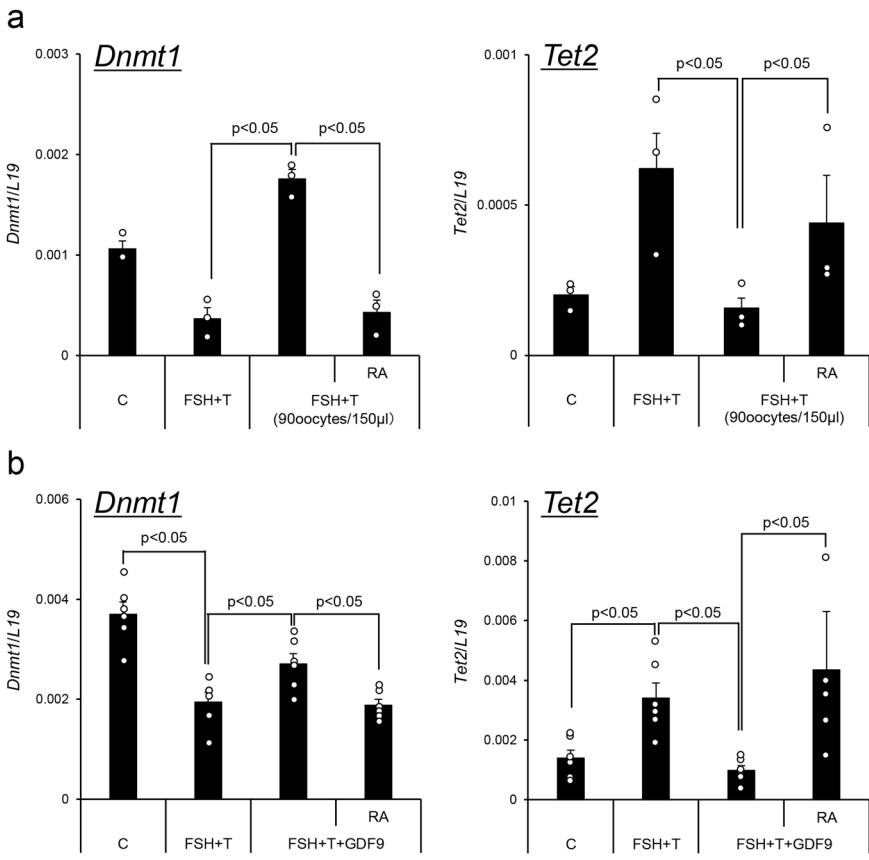

**Fig. 6 Oocytes impact the expression of *Dnmt1* and *Tet2* in granulosa cells and cumulus cells. a** The effect of coculture with denuded oocytes on the expression of *Dnmt1* and *Tet2* in granulosa cells. Granulosa cells were collected from ovaries of 3-week-old mice after treatment with eCG for 6 h and were cultured in medium supplemented with FSH plus T in the presence of 1% FBS. GV-stage oocytes were collected from COCs. None or 90 GV-stage oocytes/ 150 μl were cocultured with granulosa cells. Values are represented as the mean ± SEM ($n = 3$ biological replicates). Levels of mRNA were normalized to that of *L19*. FSH + T. Granulosa cells were cultured with FSH (100 ng/ml) and testosterone (10 ng/ml) for 48 h. RA, RA (1 μM) was added to medium in which granulosa cells were cultured with 90 GV-stage oocytes. Significant differences were observed compared with cells treated with no oocytes or cells treated with 90 oocytes per 150 μl of medium ($p < 0.05$). **b** The effects of GDF9 on the expression of *Dnmt1* and *Tet2* in granulosa cells. Granulosa cells were collected from ovaries of 3-week-old mice after treatment with eCG for 6 h and were cultured in each treatment group in the presence of 1% (v/v) FBS. control (C); granulosa cells were cultured without any hormones for 48 h. FSH + T, granulosa cells were cultured with FSH (100 ng/ml) and testosterone (10 ng/ml) for 48 h. FSH + T + GDF9, granulosa cells were cultured with FSH + T and mouse recombinant GDF9 (100 ng/ml). FSH + T + GDF9 + RA, Granulosa cells were cultured with FSH + T + GDF9 and RA (1 μM) for 48 h. Levels of mRNA were normalized to that of *L19*. Values are represented as the mean ± SEM ($n = 6$ biological replicates). Significant differences were observed among the treatment groups ($p < 0.05$).

replication sites and interacts with PCNA to ensure inheritance of the existing DNA methylation patterns at S-phase and faithful daughter cell identity[37]. During S-phase, UHRF1 and DNMT3a interact with DNMT1 on the N-terminal domain and convert unmethylated cytosine to 5-methylated cytosine[38]. When TET factors are recruited to methylated regions of DNA during S-phase, TET converts 5-methylated cytosine to 5-hydroxymethylcytosine via oxidative demethylation mechanisms, leading to demethylation

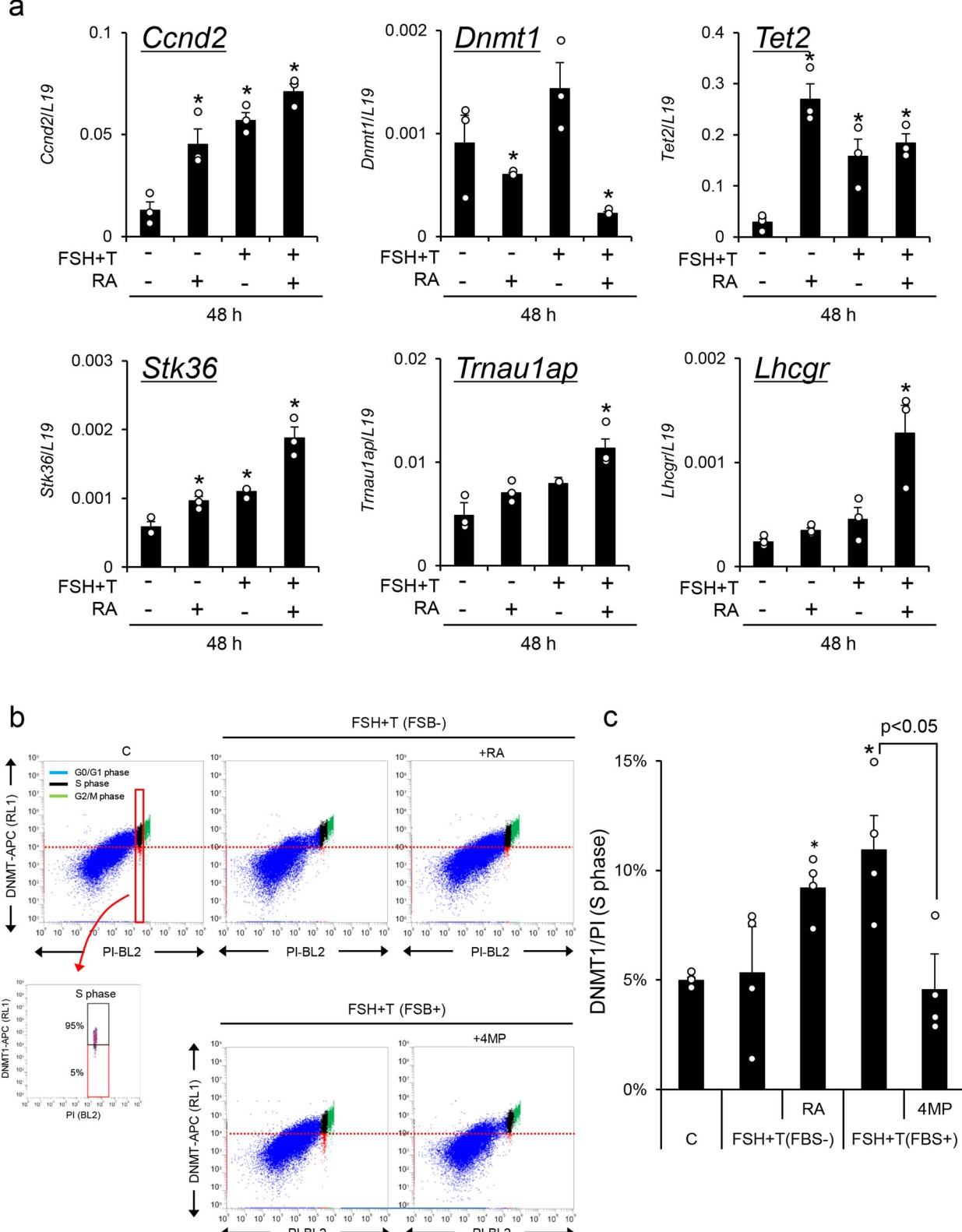

of methylated DNA regions[39]. In the present study, DNMT1 was increased in PCNA-positive S-phase granulosa cells compared with that in cells in the G0/G1 phase, thereby participating in replication accuracy. However, the level of DNMT1 in S-phase decreased in granulosa cells exposed to eCG, to induce follicular development, indicating that the DNA methylation status associated with the induction of genes controlling differentiation would be related to both the loss of the maintenance of 5-methylated cytosine due to decreased levels of DNMT1 and the conversion from 5-methylated cytosine to 5-hydroxymethylcytosine due to an increase in TET2 in S-phase. Strikingly, CpG islands in which all CpG sites were not methylated in each promoter were newly observed in granulosa cells after eCG injection in the promoter regions of *Lhcgr*, *Stk36*, and *Trnau1ap* (which included the top nine demethylated genes in

**Fig. 7 The roles of retinoic acid (RA) in epigenetic regulation in granulosa cells during follicular development. a** The expression of *Ccnd2, Dnmt1, Tet2, Lhcgr, Stk36,* and *Trnau1ap* in cultured granulosa cells. Granulosa cells were collected from ovaries of 3-week-old mice after treatment with eCG for 6 h and were cultured in the absence of serum. FSH (100 ng/ml). T, testosterone (10 ng/ml). RA (1 μM). Levels of mRNA were normalized to that of *L19*. Values are represented as the mean ± SEM ($n = 3$ biological replicates). The value of the control (without any factors) was set as 1 and the data are presented as fold induction. *Significant differences were observed between the control group and the RA and/or FSH + T treatment group ($p < 0.05$). **b, c** The percentage of cells that expressed lower levels of DNMT1 was significantly increased by RA but decreased by the alcohol dehydrogenase (ADH) inhibitor (4MP). **b** The dot plot graph of double-positive granulosa cells with DNMT1 and PI during follicular development. **c** The percentage of cells that expressed lower levels of DNMT1 at S-phase. The lower threshold of DNMT1 levels at S-phase was set as 5% of cells with lower DNMT1 expression in the control group (C; without any factors) (red line). The cells that expressed lower levels of DNMT1 are shown in the red box, and cells with higher DNMT1 expression are shown in the black box at S-phase. Values are represented as the mean ± SEM ($n = 4$ biological replicates). *Significant differences were observed between the control group and the RA treatment group in the presence of FSH + T or the FSH + T treatment group in the presence of 1% FBS ($p < 0.05$). Significant differences were observed compared with those treated with no 4MP ($p < 0.05$). Significant differences in percentage values were transformed into normally distributed numbers by angle transformation and then analyzed by one-way ANOVA. Tukey–Kramer was used as post hoc test.

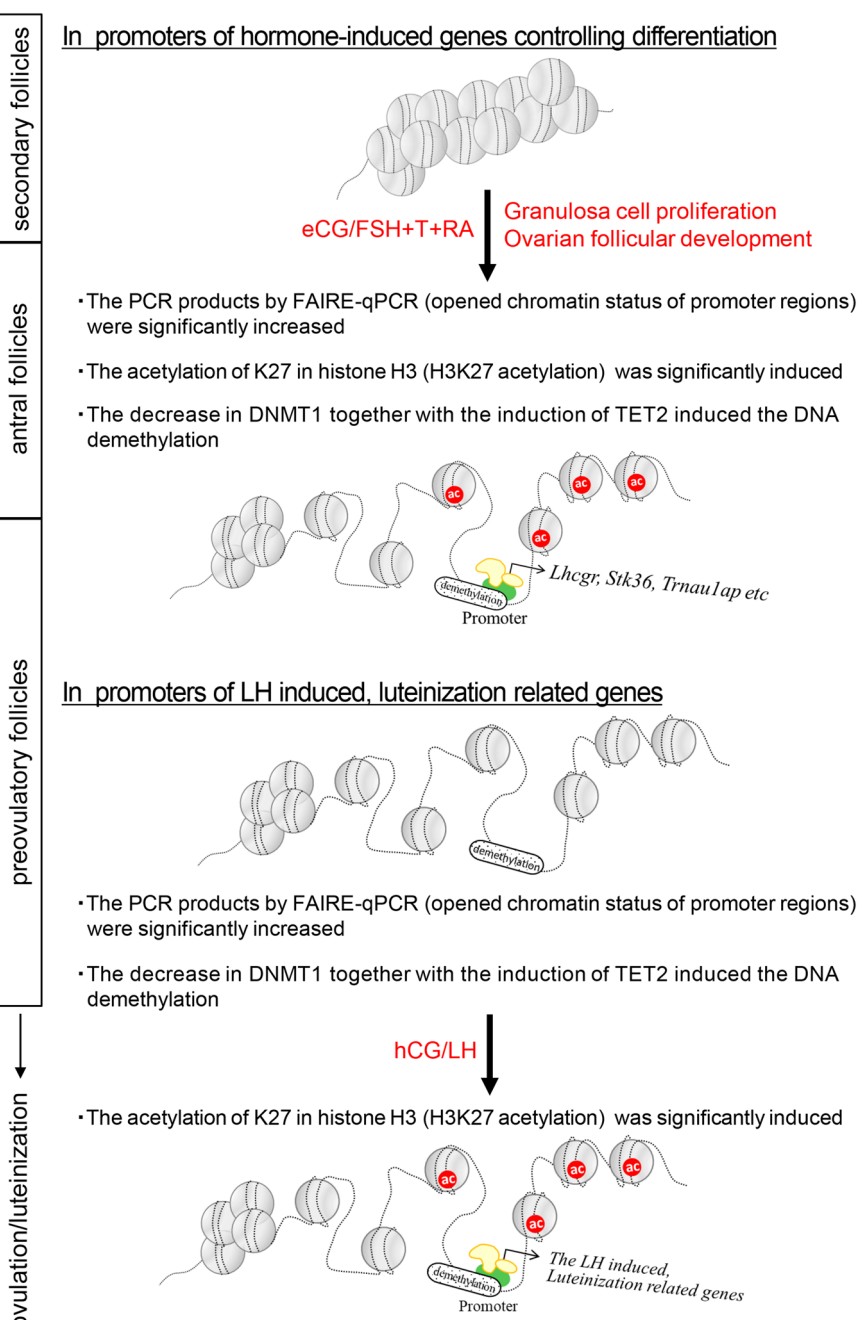

**Fig. 8 A schematic of epigenetic-related mechanisms in granulosa cells during follicular development process.** DNA demethylation and histone modification contribute to changes in gene expression as proliferating granulosa cells undergo progressive differentiation during follicular maturation.

granulosa cells). The findings supported our hypothesis that the methylation pattern would not be copied in daughter cells where the expression level of DNMT1 was low. This model by which granulosa cell functions change with proliferation is compatible with the mechanisms by which B cells and epithelial cells acquire new functions in a cell proliferation-dependent manner[20,35].

To identify the promoters of genes that were selectively demethylated in S-phase during granulosa cell differentiation, MeDIP-Seq analyses were performed. Highly demethylated promoters accounted for ~40% of the genome, including not only well-known granulosa cell markers but also promoters of other genes. Annotation analysis predicted changes in metabolic processes, phosphorylation/signaling pathways, cytoskeletal organization, transmembrane transport, catabolic processes, and chromatin modification. The factors involved in transcription and DNA modification were also significantly changed. During the follicular development process, mitochondrial ATP production is dominant and increased in granulosa cells after eCG injection, and this process is required for cell proliferation[40]. Multiple signaling pathways are activated in granulosa cells during the follicular development process, including the serine/threonine kinase or PI3K/AKT pathways[41]. In the top nine demethylated genes, *Stk36* (serine/threonine kinase 36) inhibits the activation of Gli factors that are activated by the hedgehog signal transduction cascade and regulate cell proliferation and tumorigenesis[42]. *Tranu1ap* (tRNA selenocysteine 1 associated protein 1) has been shown to inhibit proliferation in cancer cells by acting through the PI3K/AKT pathway[43]. Moreover, it is also well known that in granulosa cells, changes in the cytoskeleton and cell membrane transport occur during the follicular development process[44], suggesting that dynamic DNA demethylation in the promoter region of the genome is essential for granulosa cell proliferation and differentiation. However, the functions of most demethylated genes detected in this study, including the top nine genes, such as *Stk36* and *Trnau1ap*, remain to be determined.

Interestingly, in the genes that were induced in granulosa cells during the ovulation process (after hCG stimulation), the demethylation of promoter regions had already occurred before hCG stimulation. In the promoters of LH target genes, *Star* and *Cyp11a1*, histone-H4 acetylation (Ac-H4) and trimethylation of histone H3 lysine-4 (H3K4me3) are increased, whereas H3K9me3 and H3K27me3 are decreased in luteinized granulosa cells after hCG injection[45]. Of critical importance, granulosa cells isolated from small follicles are not capable of undergoing luteinization when cultured with LH or forskolin + phorbol 12-myristate 13-acetate (PMA), which are compounds that mimic LH signaling in preovulatory granulosa cells[46]. In contrast, granulosa cells isolated from PFs not only express high levels of the LH receptor but also rapidly differentiate into luteinized cells by both treatments[46], suggesting that histone modification is induced by the LH-stimulating signaling pathway in promoter regions where DNA demethylation and changes in DNA structure have already occurred. The histone modifications and the expression of LH target genes critical for ovulation were induced by only hCG in this study, as shown in previous reports[47]. On the other hand, in FSH-target genes that were expressed in granulosa cells by eCG injection, DNA demethylation, changes in DNA structure and histone modification were induced at the same time points. The timing of histone modification is dependent on the types of transcription factors that are activated by different signaling pathways and recruited to histone acetylation factors[48,49]. Thus, the order of DNA demethylation, changes in DNA structure, and histone acetylation appears to be strictly determined in each promoter region in granulosa cells.

Granulosa cells and cumulus cells undergo differentiation in antral follicles; however, their functional differences appear to be related, in part, to factors produced either specifically by the oocyte (GDF9 and BMP15) that activate SMAD signaling or by the granulosa cells (RA) that enhance differentiation[19,50]. Cumulus cells directly surround oocyte and transfer energy sources via gap junctional communication to oocyte during the follicular development process[51,52]. After the LH surge of ovulational stimuli, cumulus cells are not luteinized but accumulate hyaluronan-rich matrix, which impacts ovulation and fertilization processes[53]. Granulosa cells are luteinized to form the corpus luteum, which produces progesterone to induce and maintain pregnancy[54]. Herein, we show that coculture of granulosa cells with denuded oocytes maintained the expression of *Dnmt1* but suppressed the expression of *Tet2*, thus favoring promoter methylation in granulosa cells. These effects on oocytes were also induced by treatment with GDF9 and their negative effects were completely overcome by exogenous RA. RA is produced from retinol in two steps[55], and the reactions are dependent on the expression of ADH and ALDH in FSH-stimulated granulosa cells during follicular development[56]. Importantly, the induction of both enzymes is suppressed in granulosa cells by coculture with oocytes[19]. Thus, the proliferation of granulosa cells that leads to an increase in follicle diameter might be required to allow RA synthesis by reducing the local concentration of oocyte-secreted factors in follicular fluid.

In antral follicles, follicular fluid accumulates within the follicle that separates granulosa cells from the enclosed cumulus cell–oocyte complex. As cumulus cells are strongly regulated by the oocyte via oocyte-secreted factors that activate SMAD pathways[57], RA production is limited[19], and a high level of *Dnmt1* was observed in cumulus cells of PFs. Thus, the distance from the oocyte determines the epigenetic status in follicular somatic cells and their fate as cumulus cells or granulosa cells. In other words, cell proliferation first indirectly weakens the mechanisms of precise copying of the DNA methylation status due to the distance from oocyte and second, cell proliferation directly changes the methylation status and transcriptome in granulosa cells during the follicular development process. Collectively, these results indicate that the epigenetic regulation of granulosa cell differentiation mediated by cell proliferation, RA, and oocyte-secreted factors is one of the most highly orchestrated processes in female reproduction.

The maturation and developmental competence of oocytes decrease with increasing age in not only female mice but also women[58,59]. One of the reasons has been reported to be that the level of oocyte-secreted factors is lower in oocytes recovered from aged mice than in oocytes recovered from younger mice[60]. In older infertility patients, abnormal luteinization and low quality of oocytes in small antral follicles have been observed[61,62]. The decreasing ovarian functions in older women/female mice would be involved in abnormal promoter DNA methylation of critical genes in granulosa cells and cumulus cells, based on our evidence that oocyte-secreted factors strongly regulate the epigenetic changes in both cells. Moreover, bacterial infections of the female genital tract result in pelvic inflammatory disease, which causes infertility[63]. The injection of lipopolysaccharide, which is a component of gram-negative bacteria, alters the DNA methylation status in the *Lhcgr* promoter region in granulosa cells of the mouse ovary[64]. Based on the present study that focused on the changes in epigenetic status in granulosa cells during follicular development in immature mice, it is expected that analyses of the epigenetic status in granulosa cells will make it possible to identify potential causes of reproductive disorders and infertility. In particular, the analysis of DNA methylation in granulosa cells may lead to the development of new treatments and preventions for infertility or the development of contraceptives, because DNA

methylation status is an index (or measure) of whether gene expression is possible or not.

In conclusion, during follicular development, granulosa cells progressively proliferate and differentiate to acquire the ability to respond to LH, promote ovulation and undergo luteinization. Importantly, we show herein that this process involves progressive changes in the DNA methylation patterns that are regulated by the induction of TET2 and the decrease of DNMT1 in S-phase in granulosa cells of eCG-stimulated antral follicles. The induction of TET2 and the decrease of DNMT1 are regulated not only by FSH and testosterone but also by increased levels of RA signaling and decreased oocyte-secreted factors such as GDF9. Furthermore, genome-wide DNA demethylation leads to opened genome structures that, when coupled with specific histone modifications (acetylation), promote the activation of the transcription of specific genes at specific time points. We propose that the timing of sequential molecular and epigenetic events is essential for the stepwise changes in granulosa cell differentiation during follicular development and ovulation and that the RA signaling pathway plays a key role in modifying the effects of gonadotropins and oocyte-secreted factors.

## Methods

**Materials**. Dulbecco's modified Eagle medium (DMEM)/F12 and penicillin–streptomycin were purchased from Invitrogen (Carlsbad, CA, USA), fetal bovine serum was purchased from Life Technologies, Inc. (Grand Island, NY, USA), oligonucleotide poly-(dT) was purchased from Invitrogen, and AMV reverse transcriptase was purchased from Promega Corp. (Madison, WI, USA). Routine chemicals and reagents were obtained from Nacalai Chemical, Co. (Osaka, Japan) or Sigma Chemical, Co. (St. Louis, MO, USA).

**Animals**. Immature female (3-week-old) C57BL/6 mice were obtained from Charles River Laboratories Japan (Yokohama, Japan). Twenty-two-day-old female mice were injected intraperitoneally with 4 IU of eCG to stimulate follicular growth; after 48 h, they were injected with 5 IU of hCG to stimulate ovulation and luteinization. In our experiment, granulosa cells or ovaries were collected from one mouse in each treatment group for real-time PCR analysis, western blotting, and immunofluorescence staining. For low cytometry, granulosa cells were collected from two mice in each treatment group and four mice were used for bisulfite sequence assay, MeDIP-Seq analysis, chromatin immunoprecipitation (ChIP) assay, and FAIRE-qPCR analysis in each treatment group. The animals were housed under a 12 h light/12 h dark schedule in the Experimental Animal Center at Hiroshima University and provided with food and water ad libitum. The animal study was approved by the Hiroshima University Animal Committee (Permit Number: C18–34) and the mice were maintained in accordance with the Hiroshima University Guidelines for the Care and Use of Laboratory Animals.

**Isolation of granulosa cells**. SFs with multilayered granulosa cells (Type 5b[65]) were isolated from mouse (2-week-old) ovaries by puncture with a 26 G 1/2 needle under a stereomicroscope. Granulosa cells were collected from antral follicles in ovaries of immature (3-week-old) mice at 0, 12, 24, or 48 h after eCG injection or 4 or 8 h after hCG injection following 48 h eCG injection by puncture with a 26 G 1/2 needle under a stereomicroscope (Supplementary Fig. 7).

**RNA extraction and real-time PCR**. Total RNA was obtained from mouse granulosa cells of SFs with multilayered granulosa cells (Type 5b), mouse granulosa cells or cumulus-oocyte complexes (COCs) using RNAeasy Mini Kit (Qiagen Sciences, Germantown, MD, USA) according to the manufacturer's instructions. The RNA samples were treated with DNase (Qiagen) and 50 ng of total RNA was reverse transcribed in 20 μl reaction buffer. Three microliters of cDNA was used for real-time PCR. Real-time PCR was performed on a StepOne Real-Time PCR System (Applied Biosystems, Warrington, UK) with power SYBR Green PCR Master Mix (Applied Biosystems) at 60 °C for annealing temperature and 40 cycles of amplification. The primer sets are shown in Supplementary Table 5. L19 was used as a control for reaction efficiency and variations in concentrations of mRNA in the original reverse transcription (RT) reaction.

**Genomic DNA extraction**. Genomic DNA was obtained from granulosa cells of SFs with multilayered granulosa cells (Type 5b), granulosa cells at 0, 24, and 48 h after eCG injection or cultured granulosa cells of mouse ovaries using QIAamp DNA Blood Mini Kit (Qiagen Sciences) according to the manufacturer's

instructions. One milligram of genomic DNA was prepared for the bisulfite sequence assay.

**MeDIP-Seq analysis**. One microgram of DNA was used as an input sample and treated with MethylMiner Methylated DNA Enrichment kit (Invitrogen). Three biological replicates were used for each group. The DNA was sonicated by Covaris S2 (Covaris, Woburn, MA) under the following conditions (setting: value, duty cycle: 10%, intensity: 5, cycles per burst: 200, time: 2 cycles of 60 s each). The average size of the fragment after sonication was 350 bp. The fragmented DNA was dissociated into single strands, and the methylated region was enriched by the methyl-CpG binding domain of the human MBD2 protein using MethylMiner Methylated DNA Enrichment Kit (Invitrogen[66,67]). Quantification of DNA methylation at multiple false discovery rates (FDRs) was adjusted to the p-value ($p < 0.05$) cutoff. TruSeq DNA HT Sample Prep kit (Illumina, Tokyo, Japan) was used for the paired end library method. The sequence of the collected sample was decoded using an Illumina HiSeq 2000 and the methylated region was identified by MEDIPS software. AMS were used to judge the DNA methylation level of the promoter region in each gene. The prediction of gene function was analyzed using Functional Annotation Bioinformatics Microarray Analysis (https://david-d.ncifcrf.gov/tools.jsp). The accession number for all sequence data shown in this study is DNA Data Bank of Japan Sequence Read Archive: DRA010809 (http://trace.ddbj.nig.ac.jp/dra/index_e.html). The analysis of methylation of imprinted genes is shown in Supplementary Table 4.

**Bisulfite sequence assay**. One milligram of genomic DNA and 0.3 M NaOH (Nacalai) were added to the sodium bisulfite reaction mix using EpiTect Bisulfite Kit (Qiagen Sciences). Following the reaction, the bisulfite-converted DNA was purified and then used for PCR using specific primer sets to amplify the sequences containing CpG islands in the promoter region of each gene. The primer sets are shown in Supplementary Table 6. Sequence analysis of PCR products was performed on more than eight colonies for each treatment. The methylation of CpG islands in the promoter region was analyzed by QUantification tool for Methylation Analysis (http://quma.cdb.riken.jp/). The percentage of nonmethylated CpG islands in the promoter region of each gene was calculated as the percentage of colonies with all nonmethylated CpG islands in the eight colonies.

**ChIP assay**. The DNA–protein complexes were collected from SFs with multi-layered granulosa cells (Type 5b) or granulosa cells of mouse (3-week-old) ovaries at 0, 24, and 48 h after eCG injection or at 4 and 8 h after hCG injection. The ChIP assay was performed using SimpleChIP® Enzymatic Chromatin IP Kit (Magnetic Beads) (Cell Signaling, MA, USA)) according to the manufacturer's instructions. Three or four biological replicates were performed for each group. Then, 100–1000 bp of the sheared chromatin was obtained by ultrasonic disruptor (TOMY UD-200). ChIP was performed using anti-histone H3 containing the acetylated lysine 27 (H3K27ac) antibody (1 : 100) (Cell Signaling). Normal rabbit IgG antibody (1 : 100) (Cell Signaling) was used as a negative control and histone H3 (D2B12) XP® rabbit mAb (ChIP formula) (1 : 100) (Cell Signaling) was used as a positive control. Three microliters of the ChIP DNA or total input DNA (diluted to 1/50) were used for real-time PCR. Real-time PCR was performed on a StepOne Real-Time PCR System (Applied Biosystems) with power SYBR Green PCR Master Mix (Applied Biosystems) at 60 °C for annealing temperature and 40 cycles of amplification. The primer sets used for the detection of each specific region are shown in Supplementary Table 6.

**FAIRE-qPCR analysis**. SFs with multilayered granulosa cells (Type 5b) were isolated from mouse (2-week-old) ovaries, and granulosa cells were isolated from mouse (3-week-old) ovaries at 0, 24, and 48 h after eCG injection or at 4 and 8 h after hCG injection. Recovery of the genome regions of open chromatin was performed according to an established procedure[68,69]. Three microliters of the opened chromatin DNA or total input DNA (diluted to 1/10) were used for real-time PCR. Real-time PCR was performed on a StepOne Real-Time PCR System (Applied Biosystems) with power SYBR Green PCR Master Mix (Applied Biosystems) at 60 °C for annealing temperature and 40 cycles of amplification. The primer sets used for the detection of each specific region are shown in Supplementary Table 6.

**In vitro culture of granulosa cells**. Granulosa cells were collected from ovaries of immature (3-week-old) mice at 6 h following injections of eCG[19]. Cells were seeded onto serum-recoated 24- or 96-well plate. Granulosa cells were treated with 100 ng/ml FSH (NIDDK, Torrance, CA)/DMEM/F12 medium, testosterone (10 ng/ml, Sigma)/ethanol, and/or RA (1 μM, Sigma)/dimethyl sulfoxide (DMSO) in the presence or absence of 1% (v/v) fetal bovine serum. Some granulosa cells were treated with 0.05, 0.5, or 5 μM aphidicolin (Sigma)/DMSO or 100 ng/ml GDF9 (mouse recombinant GDF9, R&D Systems, Inc., Minneapolis, MN) in the presence of FSH, testosterone, and 1% (v/v) fetal bovine serum.

**Isolation and coculture of oocytes with granulosa cells**. Granulosa cells were collected from ovaries of immature (3-week-old) mice as described previously in the "Methods" section. COCs isolated from each ovarian antral follicle by needle puncture. The cumulus cells surrounding oocyte were removed by repeating pipetting using a glass pipette under the stereomicroscope. The denuded oocytes at germinal vesicle (GV) stage (GV-stage oocytes) were used for the coculture study. Granulosa cells and GV-stage oocytes were cocultured in a 96-well plate (HTS Transwell-96 Permeable Support with 8.0 μm Pore Polyester Membrane no. 3374; Corning Incorporated, Kennebunk, ME) for 48 h and treated with 100 ng/ml FSH and 10 ng/ml testosterone or 1 μM RA in the presence of 1% fetal bovine serum and 0.2 mM isobutylmethylxanthine (IBMX). IBMX was added to suppress spontaneous oocyte maturation, because GV-stage oocytes strongly suppressed *Lhcgr* expression in granulosa cells.

**Immunofluorescence staining**. Ovarian tissues were embedded in paraffin wax (ThermoFisher Scientific, Inc., Waltham, USA) after dehydration. Paraffin-embedded tissue sections (5 μm) were deparaffinized. The sections were blocked with the M.O.M kit (VECTOR) and then incubated with primary antibody (1 : 100 anti-PCNA antibody, Cell Signaling#2586S, 1 : 100 anti-DNMT1 antibody, Gene-Tex#GTX116011, 1 : 100 anti-TET2 antibody, Abcam#ab124297) in 0.3% (v/v) Triton X-100 in phosphate-buffered saline (PBS) (−) overnight at 4 °C. After washing with 0.3% (v/v) Triton X-100 in PBS (−), the ovaries were visualized with Alexa Fluor 568 goat anti-rabbit secondary antibody (1 : 100) (Sigma) or fluorescein isothiocyanate-conjugated goat anti-mouse IgG secondary antibody (1 : 100) (Sigma) and DAPI (Vector Laboratories, Inc., CA, USA). Digital images were captured using Keyence BZ-9000 microscope (Keyence, Co., Osaka, Japan).

**Western blotting**. Proteins (20 μg) were separated by SDS-polyacrylamide gel (7.5%, 10%, or 12.5%) electrophoresis and transferred to polyvinylidene difluoride membranes (GE Bioscience, USA). The membranes were blocked with 5% (w/v) bovine serum albumin (Nacalai Chemical, Co.). The blots were incubated with primary antibody (1 : 1000 anti-H3K27ac antibody, Cell Signaling#8173S, 1 : 500 anti-DNMT1 antibody, GeneTex#GTX116011, 1 : 500 anti-TET2 antibody, Cell Signaling#36449, or 1 : 1000 anti-β-actin antibody, Sigma#A5316, Supplementary Table 7) overnight at 4 °C. After washing in TBST, enhanced chemiluminescence (ECL) detection was performed by using the ECL system (GE Bioscience) and appropriately exposing the blots of DNMT1 to Fuji X-ray film (Fujifilm, Tokyo, Japan). The images for western blottings of H3K27ac and TET2 proteins were detected by ChemiDoc Touch MP imaging system (BIO-RAD). The expression of each protein was calculated by the Gel Analyzer of ImageJ (ImageJ.exe).

**Flow cytometry**. For detection of DNMT1, granulosa cells were fixed with 70% (v/v) methanol in PBS (−) for 10 min at 4 °C. After washing, these cells were treated with 0.1% (v/v) Triton X-100 in DMEM/F12 at room temperature for 15 min and then further treated with the M.O.M kit (VECTOR) in DMEM/F12 at room temperature for 30 min. The primary antibody (1 : 50 anti-DNMT1 antibody, Novus Biologicals#NB100−56519APC, Supplementary Table 7) was added at 37 °C for 30 min. For detection of PI signal, fixed granulosa cells were treated with PI staining reagents (Cellstain®-PI solution, P378, 1 : 100) at 37 °C for 30 min. The fluorescence intensity of 20,000 cell-specific events was detected using Attune NxT Acoustic Focusing Cytometer Ver 2.6 (ThermoFisher). PI fluorescence was detected by BL2 channel and different stages of the cell cycle were determined by specific pick as G0/G1, S and G2/M phases indicated by first, second, and third picks, respectively. DNMT1 fluorescence was detected by RL1 channel.

**Statistics and reproducibility**. All statistics are described in figure legends. In the comparison between the SF and the follicular follicle before eCG injection, statistical analyses of data ($n > 3$) for comparison were carried out by Student's $t$-test (Statview; Abacus Concepts, Inc., Berkeley, CA). In the comparison of multiple treatment groups, statistical analyses of data ($n > 3$) for comparison were carried out by one-way analysis of variance (ANOVA). Tukey–Kramer was used as post hoc test. In the statistical analysis of % data, percentage values were transformed into normally distributed numbers by angle transformation and then analyzed by one-way ANOVA. Tukey–Kramer was used as post hoc test. Student's $t$-test and one-way ANOVA were performed if the data were normally distributed. Data are presented as mean ± SEM ($n > 3$ biological replicates) and $p < 0.05$ was considered significant.

**Reporting summary**. Further information on research design is available in the Nature Research Reporting Summary linked to this article.

## Data availability

The authors declare that the data supporting the findings of this study are available within the paper and Supplementary Data 1, and will be available from the corresponding author upon reasonable request. The accession number for all sequence data shown in this study is DNA Data Bank of Japan (DDBJ) Sequence Read Archive: DRA010809.

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

## Acknowledgements

Ovine FSH was kindly provided by Dr. A.F. Parlow, the National Hormone and Pituitary Program, the National Institute of Diabetes and Digestive and Kidney Disease, USA. This work was supported in part by JSPS KAKENHI Grant Number JP 19H03108 (to M.S.).

## Author contributions

T.K. performed the experiments and analyzed the data. M.S. and J.S.R. designed the study, and wrote and revised the manuscript.

## Competing interests

The authors declare no competing interests.
