## [Peer Review File · Communications Biology]

Reviewers' comments:

Reviewer #1 (Remarks to the Author):

The manuscript, "Hormonally-regulated expression of DNMT1 and TET2 mediates large-scale DNA demethylation in proliferating ovarian granulosa cells (GCs)" by Kawai et al determine the underlying mechanisms by which patterns of DNA methylation change dramatically in highly proliferative GCs as they differentiate in response to key regulatory factors. They proposed that, during GTH-dependent folliculogenesis, the promoters of specific genes in granulosa cells undergo demethylation and histone acetylation under the co-regulation of FSH, RA and egg-derived factors, and they speculated that these epigenetic modifications may be required for ovulation and luteinization induced by the LH peak

On the whole, the authors have done a good deal of research. They preliminarily explored the epigenetic modification of genome during gonadotropin-induced GC proliferation and differentiation. I agree that these results are likely to be important as the foundation for further studies. However, the experimental design and results analysis of this paper are not clear enough. I have several concerns as follows.

METHODOLOGY:

1. There is a lack of information about experimental design and scheme in materials and methods.
2. There are also some ambiguities in the sampling method. For example, the secondary follicles at different stages of development have great differences in morphology and metabolic state. Therefore, the authors need to indicate which stage of secondary follicles were used in the study (Pedersen T and Peters H., 1968). In addition, how to isolate secondary follicles and how to obtain GCs from secondary follicles require detailed description.
3. Line 476, "Granulosa cells were collected from ovaries of immature mice (3 - week - old) at 6 h following injections of eCG". Why are GCs collected 6h after injection in eCG? What is the rationale for this?
4. In my opinion, neither "Student's T-test" nor "One-way ANOVA" is applicable to the significance test between percentages in the results. I suggest that authors consult an expert in statistical analysis of scientific data and that re-analyse the data.

RESULTS:

1. If I understand correctly, the author wants to study the changes in methylation in GCs under the regulation of eCG. However, in Figure 1A, it seems that the expression level of methylation-related genes in secondary follicles rather than eCG-0h is used as a comparison criterion. I think this comparison method is unreasonable. It only reflects the changes in the epigenetic modification of GCs in two developmental stages, and does not indicate that this change is caused by eCG.
2. In Figure 1C, the pictures of antral follicle presented by the author are not actually antral follicle, it can only be counted as small antral follicle at best. Small antral follicles and antral follicles are very different in morphology and metabolism. In mice, the formation of secondary follicles and small antral follicles does not depend on the induction of gonadotropins. On the contrary, antral follicles must be formed under the stimulation of gonadotropins. Therefore, the author's description of an antral follicle as an antral follicle is not rigorous.
3. L245: Can GDF9 and BMP15 inhibit the demethylation of GCs? I suggest the authors to study the methylation level or the change of related gene expression in GCs under the addition of GDF9 and BMP15.
4. L249: Please confirm "Figure 5B" or "Figure 6B"? Line 251: Similarly, where is Figure 6C? In Figure 6B, the abscissa of the histogram is not clearly labeled.
5. The data in Figure 6 and Figure 7 show that RA plays a key role in the regulation of demethylation of GCs. In the absence of FSH and T, RA can also independently regulate the demethylation of GCs. Does that mean that the demethylation of GCs induced by FSH is mainly mediated by RA? The author should add relevant experiments to verify this issue.

OTHER CONCERNS

1. The discussion consists mainly of an expanded repetition of the results. The authors should address the importance of the results from the study but not repeat the result, otherwise, I cannot comment on the scientific content of this manuscript.
2. I suggest the authors to reconsider the title. The present title do not provide enough

information about the scientific content of the present study.

3. Line 49: I think this sentence over-interprets Reference 5. There is not enough evidence that the proliferation and differentiation of GCs are closely related.

4. Line 52-53: To my knowledge there are far more than 500 target genes induced by FSH in GCs. Authors need to cite the latest references.

5. According to the data in Figure 2A, we can infer that the proliferation activity of granulosa cells reaches the highest at 48h of eCG. This does not seem to be consistent with the study proposed by another study that eCG12-24h is the period of the strongest proliferation activity of GCs (<https://doi.org/10.1093/molehr/gaaa069>). Can the author comment on this contradiction?

Reviewer #2 (Remarks to the Author):

This manuscript by Kawai et al. aims to identify what type of epigenetic modifications transpire in granulosa cells (GC) during their differentiation and proliferation from small-follicle to antral to the preovulatory stages of folliculogenesis. Here they use immature mice and treat them with eCG which exerts FSH-like actions (as mentioned by them) on GC proliferation to understand the alterations in DNA methylation profiles of FSH-induced genes and later hCG to induce ovulation to investigate the methylation changes in LH-induced genes as the GCs undergo terminal differentiation. This is an interesting area to study. However the justifications of the different experiments they carried out is not clear though they have used a good number of techniques and experiments to address their research question. However, the paper becomes extremely complicated once the angle of FSH+T+RA is introduced and therefore needs simplification. Overall it is very very difficult to understand the paper. Some key points that may be added to improve the manuscript are as follows:

General Comments:

- 1) The manuscript requires multiple reading efforts to understand the exact research question and the aim/objectives of the study. The introduction lacks clarity and focus about what lacunae in the field prompted them to initiate the study, what specific research questions are they trying to address, and what is the significance of the study.
- 2) There are grammatical and syntax errors in several places within the manuscript. Eg: line 31 (granulosa cell - not cells proliferation), line 59 (dependent, not depended), line 276 (development- not developmental related changes), line 704 (scale bar not bur). Hence the manuscript needs to be proofread thoroughly for such errors before revision.

Comments on scientific aspects:

- 1) eCG (previously known as PMSG) is known to hyperstimulate the ovarian follicles and has both FSH and LH like activities (not only FSH as mentioned by authors). But why was eCG used instead of FSH to study the FSH-dependent GC proliferation and induction of FSH targeted genes is not clear in experiments reported in Fig1, Fig.2. Further eCG, so understand FSH like activity why they used eCG? In further cell-culture experiments though, only FSH was used and not eCG. Thus, gene induction profiles upon individual use of eCG and FSH cannot be similar.
- 2) The authors mention in Materials and Methods (page15, line-528) that they used both Student's t test (possibly only for Supplemental figure 3) as well as ANOVA for statistics. However since for all statistical comparisons the analysis is between more than 2 groups, only 1-way ANOVA with a valid post-hoc test applies for the comparisons. The type of analytical and post-hoc tests used should be specified in Materials section. Also, none of the figure legends incorporate the names of the applied tests and need to be indicated appropriately.
- 3) For representing significance between the bar graphs, p values indicated by (*/**/***) provide better clarity about the difference between individual groups. The a,b system is very confusing for third-party readers. I recommend that the figures be changed to the following format to avoid confusion.

- 4) Figure 1: Authors have provided immunofluorescence figures for DNMT1/PCNA and TET2/PCNA localization in SF, A and PF. However, the same was not provided for H3K27ac or KDM6A even though the ChIP assay has been later carried out in SF and A follicles. The authors need to provide valid reasons for this or perform immunofluorescence and provide relevant data for these proteins.
- 5) In Fig. 1A, Tet1, Tet3, Kdm6b mRNA levels at 48 hrs (preovulatory follicle) appear significantly higher than SF since the data is represented as mean + SEM. Authors are requested to re-evaluate these results and provide the 1-way ANOVA data for these genes.
Fig.1C.
- 6) Figures 1B, Supplemental Figs.3E and 4A showing images for western blots of H3K27ac and TET2 proteins show a great many non-specific bands. The standardization appears sub-optimal showing much background staining of the blot. This would introduce error in densitometric quantification of the bands of interest. We recommend refining the quality of the blots for better accuracy of results.
- 7) Fig. 1C, and Supplemental Fig1A.- It is surprising that PCNA incorporation was found to be highest in follicles labelled as "preovulatory", because in preovulatory stage, the GCs reach terminal differentiation after undergoing luteinization and show the least level of proliferation. Again, the follicles labelled as "preovulatory" look more like late antral follicles, so if these results are correlated in context to late antral follicles, they make sense but not in preovulatory follicles. Also, from the pictures, follicles labelled as PF actually appear to be more like follicles of the late antral stage with oocyte in the center surrounded by a much smaller antrum compared to what is actually seen in PF or Graafian follicles. Similarly the follicles labelled as antral are more like secondary follicles in appearance where the antrum is barely visible. There is a serious observational error that may have been repeated while reporting on antral and PF follicles in all sections of the manuscript and needs to be rectified.
- 8) Page 5, Line no. 136, the subtitle is misleading. It needs to say that induction of DNMT1 is significantly decreased in S-phase granulosa cells of late antral follicles (considering that these follicles are not preovulatory) compared to smaller follicles. Also, the graph shows PI-staining whereas no mention of PI is included in the text (Pg.5 lines 136-152)
- 9) Page 5, Line no. 147, the authors mention that "induction" of DNMT1 in control GC (0hr) cells from G0/G1 to S to G2/M stages was different compared to eCG treated cells at 48 hr. However, induction is not the correct usage of word here. It is just that the DNMT1 expression is getting lowered as the follicles progress from 0hr to 48 hr from primary to late antral stages.
- 10) Fig1 has small follicles (SF) included as a control set of cells that are not treated with eCG while SFs are not included in Fig. 2. This has created much confusion for the reader while interpreting the results and the explanations for this disparity needs to be rectified.
- 11) Fig.3D. The p values should have been provided. It is difficult to appreciate the significance only by looking at the Mean & SEM values of individual methylation levels. Also the authors should mention whether the Mean SEM values from MeDIP analysis are for specific individual CpG sites or for a nucleotide stretch.
- 12) Fig. 3E should be included as a Supplemental table instead.
- 13) In S2a-c showing OCR, H3K27ac and transcript expression changes, the graphs of genes should be put in the same order as shown in S1-B.
- 14) Line 215 and Fig. S3-c. It is very abrupt and unclear as to why the AREG experiment was included to look at H3K27ac of LH target genes. If not necessary, this information needs to be omitted from the paper for simplification of results and interpretation.
- 15) Lines 226 to 237. The word alleles needs to be replaced by CpG sites. Line 235-237, the authors are making a presumptive statement by directly attributing the demethylation of FSH target genes to DNMT1 downregulation and TET2 upregulation. This sentence needs to be expressed differently eg: by using the words may be or possibly.
- 16) Line 392 – TETS do not inactivate the methyl group. It participates in demethylation of CpG and non-CpG bases via oxidative demethylation mechanisms. The sentence should be rewritten.
- 17) Lines 341-353 do not fit in the context of the paper and are just adding to the bulk in discussion. These can be removed or simplified.
- 18) Lines 354-355 – the sentence that the promoters of the three LH target genes genes are tightly packed needs to be rewritten. Consider writing hypermethylated instead. But overall the paragraph from lines 354-368 are important findings of the paper and are highly appreciated.

- 19) Lines 369-371-The statement that large scale demethylation did not occur in cumulus cells is not seen in the data provided. This sentence needs to be removed as it borders on over-interpretation of data.
- 20) Fig 6 and Fig 7 and the related results and discussion is very very complicated to go through because of the a,b style of significance shown in bar graphs and the complicated design of experiments. The reader loses focus of the manuscript here.
- 21) Overall, the paper needs to be simplified beyond the point where Fig 6 begins.

Reviewer #3 (Remarks to the Author):

In this manuscript, Kawai et al describes the mechanisms of activation of proliferation of granulosa cells. Granulosa cells are important for follicle maturation and their regulation is still not well understood. The manuscript is technically sound and have a number of interesting results. However, some data are not convincing and the description of large part of the methods section is very superficial. A better explanation of the techniques and details are required to judge the results. I believe that in present form, the manuscript is preliminary and requires improvements. My major critic is that the Methods section of the manuscript does not provide enough details so that someone could reproduce the results, and also it is difficult to interpret the results.

Major points

1. Introduction is too long and the major hypothesis is lost.
2. Methods part is not clear, the authors did not provide sufficient details of the experiments, so it is hard to interpret some data.
3. In animal study, the treatment doses, the number of animals for each treatment group, the drug dissolving and delivery, the manufacturer of the drug were not described.
4. In RNA extraction method, it is not clear how much of total RNA was used for RT QPCR, did authors treat RNA with DNase?; the use Qiagen RNeasy mini kit, does not guarantee the absence of DNA.
5. MedIP -seq, how DNA was sonicated, instrument, parameters, in which solution? What was the average size of the fragment after sonication, which antibody (reference and manufacturer) was used? How many biological replicates were used for each group? what is cut-off parameter for differential methylation analysis? Whether the multiple test (FDR) was applied, if yes, which cut-off p-value was used? If there was any internal quality control, such as analysis of methylation of imprinted genes or spike-in control used?
6. ChIP assay, how many biological replicates were used for each group, what is the size of the sheared chromatin, which negative and positive control for ChIP were used? The quantity of antibody used for ChIP and the reference and manufacturer of the antibody must be provided.
7. Figure1, WB analysis of H3K27Ac is not convincing. For histone detection by WB, the histone purification prior of WB is recommended.
8. Fig 1C, Immunofluorescence pictures, the manufacturers of the antibodies and the references must be provided. Did authors use the same exposure time for image acquisition? How many images did the authors analyze and how many replicates?
9. Figure3C, shows clusters, it will be more informative if the authors indicate, at least for some clusters, the functionally important genes.
10. It is difficult to appreciate the Figure 3E as there is no connection between the table of functional annotation and the observed effects that were discussed. The authors should discuss the detected alterations in group of genes combined by common function, eg. "Chromatin modification" and the possible impact on biological effects which were observed.
11. Discussion is very long and it is often the repetition of the Results parts. It should be reduced to a concise version.

Response to Reviewers

Reviewer #1

METHODOLOGY:

1. There is a lack of information about experimental design and scheme in materials and methods.

Thank you for your comments. To describe the aims of each experiment, we have added a paragraph that introduces our experimental design in the Materials and Methods section as follows (Page 16, line 529-Page 17, line 583).

Experimental Design

Experiment 1: The expression of genes involved in DNA methylation was analyzed in granulosa cells of secondary follicles (SF) or antral follicles (AF) before or after eCG injection as shown in Supplemental Figure 6. The expression of mRNAs was determined by real-time PCR (Figure 1A); protein levels were determined by western blotting in Figure 1B, Supplemental 5A. The immuno-localization of DNMT1 and TET2 in follicles at each stage of development is shown in Figure 1C.

Experiment 2: To determine how the levels of DNMT1 related to cell cycle progression in granulosa cells, these cells were collected from antral follicles (AF) present in mouse ovaries at 12, 24 or 48 h after eCG injection and were analyzed by FACS to determine the levels of DNMT1 and DNA (Figure 2, Supplemental Figure 2B,C).

Experiment 3: The low levels of DNMT1 associated with DNA replication in granulosa cells of preovulatory follicles suggested that promoter DNA methylation status might be limited during DNA replication. Therefore, we analyzed the DNA methylation status of granulosa cells during follicle development at the whole genome level. The analysis of genome-wide DNA methylation status in granulosa cells during follicular development is shown in Figure 3 and Supplemental Table 1-4. Validation of DNA methylation status by bisulfite sequence assay was done for 3 genes (Stk36, Trna1ap and Lhcgr) selected from top nine most highly demethylated genes (Figure 4A). To examine the impact of DNA demethylation on gene expression, open chromatin analysis by FAIRE-qPCR analyses (Figure 4B, Supplemental Figure 3A), chromatin immunoprecipitation assays of acetylated H3K27 (Figure 4C, Supplemental Figure 3B) and gene expression analyses by Real-time PCR analysis (Figure 4D, Supplemental Figure 3C) were done in the top nine most highly demethylated genes.

Experiment 4: Based on the results of MeDIP-Seq analyses, the promoter regions of LH target genes (Star, Cyp11a1 and Ptgs2) were also significantly demethylated in granulosa cells in response to eCG stimulation. Therefore, to determine the mechanisms by which the LH target genes were expressed after, but not before, hCG injections, FAIRE-qPCR analyses were done to determine the chromatin structure of specific promoter regions (Supplemental Figure 4A), CHIP assays using anti-H3K27 acetylated histone H3 (Supplemental Figure 4B) and Real-time PCR analyses (Supplemental Figure 4C) were performed using granulosa cells isolated at 48 hours after eCG injection (0h hCG) and granulosa cells at 4 or 8 h after hCG injection.

Experiment 5: The DNA methylation status in genome-wide promoter regions dramatically changed in proliferating granulosa cells after eCG injection by suppressing expression of DNMT1 and increasing that of TET2. Therefore, to understand the relationship between cell proliferation and demethylation, the pattern of demethylated CpG sites in each promoter region was analyzed by bisulfite sequence analysis (Figure 5A). Using the in vitro culture model to mimic follicular development, the effects of a cell cycle inhibitor (aphidicolin) on the demethylation in the promoter regions (Figure 5B) and on gene expression (Figure 5C) were examined. Granulosa cells were cultured with FSH and testosterone (FSH+T) and/or aphidicolin (aphi) for 48 h.

Experiment 6: Retinoic acid (RA) is one of the key inducers of Lhcgr expression in granulosa cells and the co-culture with oocytes suppresses the response to RA. Therefore, to determine the roles of RA and oocyte-secreted factors in DNA demethylation in granulosa cells during follicular development, granulosa cells were cultured with oocytes and/or RA in the presence of FSH+T. The expression levels of Dnmt1 and Tet2 were examined (Figure 6). Furthermore, granulosa cells were cultured with or without FSH+T and/or RA under the absence of FBS because FBS contains RA precursor retinol. The expression of specific genes, Ccnd2, Dnmt1, Tet2, Stk36, Trna1ap and Lhcgr was examined (Figure 7A). Lastly, granulosa cells were cultured with FSH+T and/or RA in the absence of FBS or cultured with FSH+T and/or 4MP (an ADH inhibitor that suppresses the conversion of retinol to RA) in the presence of FBS. The granulosa cells were used for FACS analysis to detect the levels of DNMT1 and DNA in granulosa cells exposed to each hormone and agonist/antagonist regimen (Figure 6B).

2. There are also some ambiguities in the sampling method. For example, the secondary follicles at different stages of development have great differences in morphology and metabolic state. Therefore, the authors need to indicate which stage of secondary follicles were used in the study (Pedersen T and Peters H., 1968). In addition, how to isolate secondary follicles and how to obtain GCs from secondary follicles require detailed description.

We have added information about which stage of secondary follicles was used in this study, how to isolate the secondary follicles and how to obtain granulosa cells from the secondary follicles in Material methods as follows according to the paper by Pedersen T and Peters H., 1968.

Page11, line387-392

Isolation of granulosa cells

Secondary follicles with multilayered granulosa cells (Type 5b [57]) were isolated from mouse (2-week-old) ovaries by the puncture with 26G 1/2 needle using a stereo microscope. Granulosa cells were collected from antral follicles in ovaries of immature (3-week-old) mice at 0, 12, 24 or 48 h after eCG injection or 4 or 8 h after hCG injection following 48-h eCG injection by the puncture with 26G 1/2 needle using a stereo microscope (Supplemental Figure 6).

3. Line 476, "Granulosa cells were collected from ovaries of immature mice (3-week-old) at 6 h following injections of eCG". Why are GCs collected 6 h after injection in eCG? What is the rationale for this?

In preliminary experiments, the response to FSH was highest in granulosa cells collected from immature mice 6 h after eCG injection, as compared with that in granulosa cells of immature female mice without any hormonal treatment. Moreover, spontaneous luteinization was not induced when granulosa cells were recovered and cultured 6 h after eCG injection. Therefore, we used the *in vitro* culture condition to examine the specific effects of FSH alone on granulosa cells functions not only in the present study but also in our previous study (Kawai et al., 2018, Endocrinology).

4. In my opinion, neither "Student's T-test" nor "One-way ANOVA" is applicable to the significance test between percentages in the results. I suggest that authors consult an expert in statistical analysis of scientific data and that re-analyses the data.

Thank you for your suggestion. We have redone the statistical analysis % data. Significant differences in percentage values were transformed into normally distributed numbers by Angle transformation and then analyzed by one-way ANOVA. Tukey-Kramer was used as a post-hoc test.

We also changed to the method of analyzing the raw data of MeIP-seq. Specifically, the data in Figure3D, Supplemental Table 1,2 were changed from percentage to Absolute methylation signals (AMS) and then analyzed by one-way ANOVA. Tukey-Kramer was used as post-hoc test.

RESULTS:

1. If I understand correctly, the author wants to study the changes in methylation in GCs under the regulation of eCG. However, in Figure 1A, it seems that the expression level of methylation-related genes in secondary follicles rather than eCG-0h is used as a comparison criterion. I think this comparison method is unreasonable. It only reflects the changes in the epigenetic modification of GCs in two developmental stages and does not indicate that this change is caused by eCG.

Thank you for your suggestion. We agree with your suggestions. We have redone the statistical analyses to compare results obtained in secondary versus antral follicles (before eCG injection). The difference is shown by p value. A separate statistical analysis was also done to determine whether there was a significant change in the expression levels of genes before and after eCG injection. These are shown in Figure 1A, Figure 4, Supplemental Figure 3. As shown, the expression of some of genes was changed in granulosa cells isolated secondary follicles versus antral follicles; however, the expression of most of genes analyzed, including that of *Dnmt1*, was changed by eCG injection. Detailed information about the statistical analyses was included in each figure legend.

2. In Figure 1C, the pictures of antral follicles presented by the author are not actually antral follicles, it can only be counted as small antral follicle at best. Small antral follicles and antral follicles are very different in morphology and metabolism. In mice, the formation of secondary follicles and small antral follicles does not depend on the induction of gonadotropins. On the contrary, antral follicles must be formed under the stimulation of gonadotropins. Therefore, the author's description of an antral follicle as an antral follicle is not rigorous.

Thank you for your suggestion. We did an additional immunofluorescence study using ovaries of immature mice to analyze secondary follicles and antral follicles, and using immature mice 48 h after eCG injection analyzing preovulatory follicles. Please see the new Figure 1C.

3. L245: Can GDF9 and BMP15 inhibit the demethylation of GCs? I suggest the authors to study the methylation level or the change of related gene expression in GCs under the addition of GDF9 and BMP15.

In our previous study (Kawai et al., 2018), we showed that the demethylation of the *Lhcgr* promoter region and *Lhcgr* expression were significantly suppressed in granulosa cells co-cultured with denuded oocytes. Moreover, induction of *Lhcgr* mRNA did not occur in cumulus cells when

intact cumulus oocyte complexes (COCs) were cultured with FSH alone. However, when intact COCs were treated with a SMAD inhibitor, *Lhcgr* mRNA was induced in cumulus cells. Therefore, our evidence strongly suggested that oocyte secreted factors (most likely GDF9 and BMP15) that activate the SMAD pathway to maintain DNMT1 expression and to retain the DNA methylation status of *Lhcgr*. GDF9 and BMP15 are well known oocyte secreted factors that activate the intracellular SMAD pathway in cumulus cells to suppress *Lhcgr* expression. However, in this study, we did not try to analyze the effects of BMP15 or GDF9 on DNA methylation status in granulosa cells. Moreover, the main aim of this study was to determine the mechanism by which DNA demethylation in genome-wide promoter regions is induced. This study shows that the reduction of DNMT1 by retinoic acid is involved in decreasing DNA methylation in granulosa cells of preovulatory follicles. Therefore, we deleted the description of BMP15 and GDF9 in the sentence as follows. Additionally, according to other reviewer's suggestions, the results that SMAD inhibitor decreased *Dnmt1* expression and increased *Tet2* in cumulus cells of COCs is similar to those by RA were deleted from Figure 6.

Page 7, Line 233-236

However, when granulosa cells were co-cultured with denuded oocytes, the effects of FSH+T were dramatically reduced (Figure 6) whereas the addition of RA to the medium overcame the negative effects mediated by the denuded oocytes (Figure 6).

In the discussion section, the term of oocyte secreted factors was only used in sentences with appropriate citations. The term was deleted in the sentences referring to the results of this study as follows.

Page 10, line 332-Page 11, 349

Granulosa cells and cumulus cells undergo differentiation during antral follicle development; however, their functional differences appear to be related, in part, to factors produced either specifically by the oocyte (GDF9 and BMP15) that activate SMAD signaling or by the granulosa cells (RA) that enhance differentiation [25,52]. Cumulus cells directly surround oocyte and transfer energy sources via gap junctional communication to oocyte during follicular development [53,54]. After the ovulatory surge of LH, cumulus cells do not luteinize but produce a hyaluronan rich matrix that impacts ovulation and fertilization [55]. By contrast, granulosa cells do undergo luteinization to form corpora lutea that produce progesterone required to induce and maintain pregnancy [56]. Herein, we show that co-culture of granulosa cells with denuded oocytes maintained the expression of *Dnmt1* and suppressed the expression of *Tet2*, thus favoring promoter methylation in granulosa cells. These effects of oocytes were completely overcome by the exogenous RA. However, RA alone

only reduced the level of *Dnmt1* but did not significantly change the DNA methylation status of the genes, including *Stk36*, *Trna1ap* in this study and *Lhcgr* shown in our previous study [25]. Combined treatments of FSH+T with RA were required for promoter demethylation and expression of these genes, *Stk36*, *Trna1ap* and *Lhcgr*. Collectively, these results indicate that the epigenetic regulation in ovarian granulosa cells by the cell proliferation, RA and oocyte-secreted factors could be one of the most highly orchestrated processes in female reproduction.

4. L249: Please confirm "Figure 5B" or "Figure 6B"? Line 251: Similarly, where is Figure 6C? In Figure 6B, the abscissa of the histogram is not clearly labeled.

Thank you for your suggestion. According to other reviewer's suggestions, we deleted Figure 6B. We carefully checked whether the number of each Figure was correct in result section.

5. The data in Figure 6 and Figure 7 show that RA plays a key role in the regulation of demethylation of GCs. In the absence of FSH and T, RA can also independently regulate the demethylation of GCs. Does that mean that the demethylation of GCs induced by FSH is mainly mediated by RA? The author should add relevant experiments to verify this issue.

Thank you for your suggestion. We did the additional study to examine whether RA alone can induce not only the changes of *Dnmt1* expression but also induce the demethylation of *Stk36* and *Trna1ap* promoter regions. As compared with control (without any stimulations), the treatment with RA alone significantly decreased *Dnmt1* expression and increased *Tet2* expression. The DNA methylation ratio in CpG islands in the promoter regions was slightly but not significantly decreased; however, further significant reduction was induced by the addition of FSH+T to RA containing medium. On the other hand, FSH+T alone (without RA) did not induce the suppression of *Dnmt1* and demethylation when the granulosa cells were cultured without FBS. Granulosa cell proliferation was not induced by the treatment with RA alone; FSH+T stimulation was required for the proliferation. Thus, RA itself can suppress the induction of DNMT1 at S phase of granulosa cell proliferation; however, demethylation was induced by both the low level of DNMT1 induced by RA and cell proliferation induced by FSH+T. Thus, the induction of gene expression depends not only on the changes in DNA structure but also requires specific transcription factors. FSH signaling might activate the signaling pathway required to strongly induce gene expression. These data are presented in Figure 7 A and supplemental Figure5. In the discussion section, the role of RA was discussed as follows.

However, RA alone only reduced the level of Dnmt1 but did not significantly change the DNA methylation status of the genes, including *Stk36*, *Trna1ap* in this study and *Lhcgr* shown in our previous study [25]. Combined treatments of FSH+T with RA were required for promoter demethylation and expression of these genes, *Stk36*, *Trna1ap* and *Lhcgr*. Collectively, these results indicate that the epigenetic regulation in ovarian granulosa cells by the cell proliferation, RA and oocyte-secreted factors could be one of the most highly orchestrated processes in female reproduction.

OTHER CONCERNS

1. The discussion consists mainly of an expanded repetition of the results. The authors should address the importance of the results from the study but not repeat the result, otherwise, I cannot comment on the scientific content of this manuscript.

Thank you for your suggestion. We reconstructed the Discussion parts, specially deleted an expanded repetition of the results.

2. I suggest the authors to reconsider the title. The present title does not provide enough information about the scientific content of the present study.

Thank you for your suggestion. We changed the title to “Large-scale DNA demethylation in proliferating granulosa cells induces progressive differentiation”.

3. Line 49: I think this sentence over-interprets Reference 5. There is not enough evidence that the proliferation and differentiation of GCs are closely related.

We agree with your suggestion; the paper about *Ccnd2* KO mice published in Nature is not enough evidence. Therefore, we carefully explained that granulosa cells proliferate, and after they proliferate, gene expression patterns change allowing responses to ovulation stimulation occurs as follows.

Page 3, line 69-79

During follicular development, granulosa cells that are essential supporters/regulators of oocyte growth and maturation also exhibit increased proliferative activity [17]. Granulosa cell proliferation is dependent on increased expression of the cell cycle regulator cyclin D2 (CCND2) [18,19]; mutant female mice lacking functional CCND2 are infertile and have a reduced number of granulosa cells in which abnormal gene expression patterns are observed [20]. In response to FSH, proliferating

granulosa cells also exhibit distinct sequential changes in gene expression patterns, involving as many as 500 genes, during each stage of follicular development [21,22]. Some of the critical genes, including the LH receptor (Lhcgr), that are obligatory for LH-induced ovulation are only expressed in FSH-stimulated granulosa cells in ovarian follicles that have reached a maximum diameter (preovulatory follicles) [23,24], suggesting that cell proliferation itself might impact granulosa cell differentiation.

4. Line 52-53: To my knowledge there are far more than 500 target genes induced by FSH in GCs. Authors need to cite the latest references.

Thank you for your suggestion. We added the references to show the dynamic changes of gene expression pattern in granulosa cells by FSH during follicular development process.

21. McRae RS, Johnston HM, Mihm M, O'Shaughnessy PJ. Changes in Mouse Granulosa Cell Gene Expression during Early Luteinization. *Endocrinology*, 2005 146: 309-317.

22. Wigglesworth K, Lee KB, Emori C, Sugiura K, Eppig JJ. Transcriptomic diversification of developing cumulus and mural granulosa cells in mouse ovarian follicles. *Biology of Reproduction*. 2015 92: 1-14.

5. According to the data in Figure 2A, we can infer that the proliferation activity of granulosa cells reaches the highest at 48h of eCG. This does not seem to be consistent with the study proposed by another study that eCG12-24h is the period of the strongest proliferation activity of GCs (<https://doi.org/10.1093/molehr/gaaa069>). Can the author comment on this contradiction?

Many studies have shown that granulosa cell proliferation is rapidly increased by FSH (eCG injection). Thus, according to your suggestions, we did an additional study to analyze the ratio of each cell cycle stage in granulosa cells collected from mice 12 h after eCG injection. The ratio of S phase was increased at 12 h after eCG injection as reported by others. However, at 12 h after eCG injection, the expression level of DNMT1 in S-phase granulosa cells was not significantly decreased. Moreover, our previous paper (Fan et al., *Science*, 2009) shows that granulosa cell proliferative activity is highest in preovulatory follicles immediately before hCG injection. Therefore, at the latter stage of follicular development process (preovulatory follicles), the reduction of DNMT1 occurred at S-phase when it impacted the epigenetic regulation of genes expressed in granulosa cells. The data about cell cycle at 12 h was included in a new Figure 2.

Reviewer #2

General Comments:

1) The manuscript requires multiple reading efforts to understand the exact research question and the aim/objectives of the study. The introduction lacks clarity and focus about what lacunae in the field prompted them to initiate the study, what specific research questions are they trying to address, and what is the significance of the study.

To clearly identify the major aims of this study we have revised the introduction. The hypothesis upon which the study was based is that hCG induced differentiation of granulosa cells in preovulatory follicles leads to the induction of the LH receptor (LHCGR), a marker of granulosa cell differentiation at this specific stage of follicle development. Specifically, the induction of the *Lhcgr* gene is dependent on a large-scale change in the DNA methylation of the *Lhcgr* promoter that would be mediated by both a decrease in the amount of the DNA methyltransferase DNMT1 and cell division occurring at the same time and that demethylation of the *Lhcgr* promoter region only occurs in a retinoic acid-dependent manner at a specific stage of granulosa cell differentiation. Therefore, the aims of this study were to determine the functional relationships among cell division, the levels of DNMT1, retinoic acid activation of *Lhcgr* transcription and epigenetic modifications of granulosa cells.

Please read our new introduction section (Page 2, line 40-Page 4, line 97).

2) There are grammatical and syntax errors in several places within the manuscript. Eg: line 31 (granulosa cell - not cells proliferation), line 59 (dependent, not depended), line 276 (development-not developmental related changes), line 704 (scale bar not bur). Hence the manuscript needs to be proofread thoroughly for such errors before revision.

We corrected the grammatical errors in all these places according to your suggestions.

Comments on scientific aspects:

1) eCG (previously known as PMSG) is known to hyperstimulate ovarian follicles and has both FSH and LH like activities (not only FSH as mentioned by authors). But why was eCG used instead of FSH to study the FSH-dependent GC proliferation and induction of FSH targeted genes is not clear in experiments reported in Fig1, Fig.2. Further eCG, so understand FSH like activity why they used eCG? In further cell-culture experiments though, only FSH was used and not eCG. Thus, gene induction profiles upon individual use of eCG and FSH cannot be similar.

eCG is used routinely (perhaps universally) by most investigators to stimulate ovarian follicular

development *in vivo* in many species (including mice) because it has both LH and FSH like activity and leads to the development of multiple preovulatory follicles (superovulation). As a consequence its LH-like activity mimics LH action on theca cells to synthesize testosterone during follicular development. The FSH-like activity mimics FSH action on granulosa cells to induce the expression of the enzyme aromatase that converts testosterone to estradiol, a key follicular steroid hormone. Testosterone also acts directly via androgen receptors to induce gene expression in granulosa cells and theca cells. To stimulate granulosa cell differentiation and the induction of *Lhcgr* expression in culture not only FSH but also testosterone or estrogen are added. Thus, we used eCG to induce follicular development *in vivo*, and FSH and testosterone to study granulosa cells differentiation in culture. We have modified the text from "FSH-stimulated granulosa cells" and "eCG-stimulated granulosa cells" to "during eCG-induced follicle development, and FSH stimulated granulosa cells". We have also stated the reason for using eCG *in vivo* in first paragraph of result section.

Page 2, line 29-32

In response to equine chorionic gonadotropin (eCG) stimulation of follicular development, granulosa cell proliferation increased, DNA methyl transferase (DNMT1) significantly decreased and Tet methylcytosine dioxygenase2 (TET2) significantly increased in S phase granulosa cells.

Page 4, line 102-105

During the induction of preovulatory follicle development by eCG, the expression levels of *Dnmt1* mRNA decreased significantly in granulosa cells isolated from mouse ovaries 48 h after eCG injection as compared with *Dnmt1* mRNA levels in antral follicles isolated mice before eCG injection (0h).

Page 7, line 218-220

When granulosa cells were collected from immature mice and cultured with FSH and testosterone, known inducers of granulosa cell differentiation *in vitro* [25], an increase of non-methylated CpG islands was also observed (Figure 5B).

Page 8, Line 280-281

However, the level of DNMT1 in S phase decreased in the granulosa cells that responded to eCG induction of follicular development.

Page 10, line 352-354

Importantly, we show herein that this involves progressive changes in the DNA methylation patterns that are regulated by the induction of TET2 and the decrease of DNMT1 in S phase granulosa cells of eCG-stimulated antral follicles.

2) The authors mention in Materials and Methods (page15, line-528) that they used both Student's t test (possibly only for Supplemental figure 3) as well as ANOVA for statistics. However, since for all statistical comparisons the analysis is between more than 2 groups, only 1-way ANOVA with a valid post-hoc test applies for the comparisons. The type of analytical and post-hoc tests used should be specified in Materials section. Also, none of the figure legends incorporate the names of the applied tests and need be indicated appropriately.

Thank you for your suggestion. We have redone the statistical analyses to compare results obtained between secondary follicles and antral follicles (before eCG injection). The statistical analyses of data from three or four replicates for comparison were carried out by Student's t-test if the data were normally distributed. The statistical analysis to determine whether there was a significant change in the expression level after eCG injection as compared with that before eCG was also done by one-way ANOVA if the data were normally distributed. Tukey-Kramer was used as post-hoc test. In comparisons of other multi-treatment groups, statistical analyses were also carried out by one-way ANOVA. Tukey-Kramer was used as post-hoc test. The detailed information about statistical analyses is described in the Materials and Methods section.

Page 15, line 520-527

In the comparison between the secondary follicle and the follicular before eCG injection, statistical analyses of data from three or four replicates for comparison were carried out by Student's t-test (Statview; Abacus Concepts, Inc., Berkeley, CA). In comparisons of multi-treatment groups, statistical analyses of data from three or four replicates for comparison were carried out by one-way ANOVA. Tukey-Kramer was used as post-hoc test. In the statistical analysis % data, percentage values were transformed into normally distributed numbers by Angle transformation and then analyzed by one-way ANOVA. Tukey-Kramer was used as post-hoc test. Student's t-test and one-way ANOVA were done if the data were normally distributed.

3) For representing significance between the bar graphs, p-values indicated by (**/****) provide better clarity about the difference between individual groups. The a,b system is very confusing for third-party readers. I recommend that the figures be changed to the following format to avoid confusion.

According to your suggestion, for representing significance between the bar graphs, we have replaced the a,b, system with the (**/**) system for all data.

4) Figure 1: Authors have provided immunofluorescence figures for DNMT1/PCNA and TET2/PCNA localization in SF, AF and PF. However, the same was not provided for H3K27ac or KDM6A even though the ChIP assay has been later carried out in SF and AF follicles. The authors need to provide valid reasons for this or perform immunofluorescence and provide relevant data for these proteins.

The main aim of this study was to determine if the reduction of DNMT1 and induction of TET2 would induce dynamic changes of DNA methylation status that would impact granulosa cell differentiation. We recognize based on your comments and those of the other reviewer that including factors involved in epigenetic regulation, such as H3K27 acetylated histone H3 and KDM6A, detracts from our focus on DNA methylation and is confusing. Therefore, we have deleted the Western blot data for H3K27 acetylated histone and mRNA level for *Kdm6a* from Figure 1. The Western blot data for H3K27 acetylated histone is now shown in Supplemental Figure 4 where Chip assay data is also shown.

5) In Fig. 1A, Tet1, Tet3, Kdm6b mRNA levels at 48 h (preovulatory follicle) appear significantly higher than SF since the data is represented as mean + SEM. Authors are requested to re-evaluate these results and provide the 1-way ANOVA data for these genes.

We agree. Therefore, in Figure 1A, we have redone the statistical analyses to compare data from secondary follicles and antral follicles (before eCG injection). The difference was shown by p-value.

For *Dnmt3* and *Tet2*, significant differences were observed between SF and eCG 0h antral follicle. The statistical analyses to determine whether there was a significant change in the expression levels of these genes after eCG injection as compared with that in before eCG injection were also done. These data were analyzed by one-way ANOVA. Tukey-Kramer was used as post-hoc test. Significant differences were observed between eCG 0h and 48h in *Dnmt1* and *Tet2*. Information about the statistical analyses that were used are included in Figure Legend. The results were shown as follows.

Page 4, line 105-109

The expression of *Dnmt3l* mRNA was significantly lower in granulosa cells present in antral follicles isolated from mouse ovaries before eCG injection as compared with those in secondary

follicles (SF). By contrast, Tet2 mRNA levels in granulosa cells increased significantly and progressively from the secondary follicle (SF) stage to the antral follicle stage prior eCG and in preovulatory follicles at 48 h after eCG injection (Figure 1A).

6) Figures 1B, Supplemental Figs.3E and 4A showing images for western blots of H3K27ac and TET2 proteins show a great many non-specific bands. The standardization appears sub-optimal showing much background staining of the blot. This would introduce error in densitometric quantification of the bands of interest. We recommend refining the quality of the blots for better accuracy of results.

We regret that the quality of western blotting was low. For TET2, we purchased a new antibody (Cell signaling (36449)) with which we were able to obtain an image with clear positive signals and fewer non-specific bands. Therefore, we repeated the experiment three times using this new antibody. By extending the sonication time, we were able to obtain a sharp band for histone H3 and repeated the experiment three times. These results are shown in Figure 1B and Supplemental Figure 4D.

7) Fig. 1C, and Supplemental Fig1A.- It is surprising that PCNA incorporation was found to be highest in follicles labelled as “preovulatory”, because in preovulatory stage, the GCs reach terminal differentiation after undergoing luteinization and show the least level of proliferation. Again, the follicles labelled as “preovulatory” look more like late antral follicles, so if these results are correlated in context to late antral follicles, they make sense but not in preovulatory follicles. Also, from the pictures, follicles labelled as PF actually appear to be more like follicles of the late antral stage with oocyte in the center surrounded by a much smaller antrum compared to what is actually seen in PF or Graafian follicles. Similarly, the follicles labelled as antral are more like secondary follicles in appearance where the antrum is barely visible. There is a serious observational error that may have been repeated while reporting on antral and PF follicles in all sections of the manuscript and needs to be rectified.

Granulosa cells of preovulatory follicles differentiate to express LHCGR and hence acquire the ability to respond to LH/hCG and ovulate and undergo terminal differentiation to luteinized, non-dividing cells. However, the preovulatory follicle has not undergone terminal differentiation and the granulosa cells are still proliferative. The LH surge expresses the factors that suppress the cell cycle as shown in our past study (Fan et al., 2009). To explain the confusion, we have added an image diagram showing when the sample was collected (Supplemental Figure 6).

8) Page 5, Line no. 136, the subtitle is misleading. It needs to say that induction of DNMT1 is significantly decreased in S-phase granulosa cells of late antral follicles (considering that these follicles are not preovulatory) compared to smaller follicles. Also, the graph shows PI-staining whereas no mention of PI is included in the text (Pg.5 lines 136-152).

Thank you for your suggestion. According to your suggestion, we now state that the induction of DNMT1 is significantly decreased in S-phase granulosa cells of preovulatory follicles. We also now mentioned the PI staining.

Page 5 line135-138

To determine the relationship between cell cycle progression and the expression level of DNMT1 in more detail, the stage of the cell cycle determined by PI staining and the expression level of DNMT1 were analyzed by flow cytometry in granulosa cells collected at specific time points.

Page 5 line144-146

However, the fluorescence intensity of DNMT1 significantly decreased in S phase-granulosa cells collected from large antral follicles (preovulatory follicles) at 48 h after eCG injection (Figure 2B).

9) Page 5, Line no. 147, the authors mention that “induction” of DNMT1 in control GC (0hr) cells from G0/G1 to S to G2/M stages was different compared to eCG treated cells at 48 hr. However, induction is not the correct usage of word here. It is just that the DNMT1 expression is getting lowered as the follicles progress from 0hr to 48 hr from primary to late antral stages.

According to your suggestion, the expression "induction was reduced" has been deleted not only in the result section but also in the discussion section. The revised sentence indicates that “the amount of DNMT1 decreased in the S phase”.

Page 5 Line144-146

However, the fluorescence intensity of DNMT1 significantly decreased in S phase-granulosa cells collected from large antral follicles (preovulatory follicles) at 48 h after eCG injection (Figure 2B).

Page 9, line280-284

However, the level of DNMT1 in S phase decreased in the granulosa cells that responded to

eCG induction of follicular development. This observation indicates that the DNA methylation status associated with the induction of genes controlling differentiation could be related to both the loss of 5mC maintenance due to decreased levels of DNMT1 and/or by the conversion from 5mC to 5hmC due to an increase of TET2 in S phase.

Page 11, line352-354

Importantly, we show herein that this involves progressive changes in the DNA methylation patterns that are regulated by the induction of TET2 and the decrease of DNMT1 in S phase granulosa cells of eCG-stimulated antral follicles.

10) Fig1 has small follicles (SF) included as a control set of cells that are not treated with eCG while SFs are not included in Fig. 2. This has created much confusion for the reader while interpreting the results and the explanations for this disparity needs to be rectified.

In Figure 1, the significant decrease of DNMT1 was observed in antral follicles after eCG stimulation. The number of DNMT1 and PCNA double positive cells decreased further during the transition of antral follicles to preovulatory follicles. SFs were not included in Figure 2 due to number of SF required to run the FACS analyses. Therefore, because a dynamic change of DNMT1 expression was observed in granulosa cells following eCG injection and because more granulosa cells could be obtained in the larger follicles, we collected granulosa cells from antral follicles/preovulatory follicles before or 12, 24 or 48 h after eCG injection and then used for FACS analyses to determine the level of DNMT1 at each stage of the cell cycle.

11) Fig.3D. The p-values should have been provided. It is difficult to appreciate the significance only by looking at the Mean & SEM values of individual methylation levels. Also, the authors should mention whether the Mean SEM values from MeDIP analysis are for specific individual CpG sites or for a nucleotide stretch.

In previous version of Figure 3D, the average value of absolute methylation signals (AMS) in SF was set as 100 % and the reduction or induction rate during follicular development was calculated. However, due to the conversion, the data no longer had a normal distribution and therefore it was impossible to perform ANOVA analysis. Therefore, in Figure 3D, the mean and SEM of absolute methylation signals (AMS) were shown. The mean and SEM values of AMS from MeDIP seq analyses showed the copy number of specific promoter region containing CpG sites in each gene that were precipitated by methylated DNA-binding protein. Statistical analysis of data from three replicates for comparison were carried out by one-way ANOVA.

Tukey-Kramer was used as post-hoc test. As compared with SF, the significant difference ($p < 0.05$) was shown as * in Figure 3D.

Page 24, Line 822-827

The mean \pm SEM values from MeDIP seq analysis are absolute methylation signals (AMS) in specific promoter regions containing CpG sites. For genes with multiple specific promoter regions including CpG sites, the average value was calculated. *; The significant differences between values in in antral follicles (3-week mice not treated with eCG) or preovulatory follicles (at 48 h after eCG injection) and that in granulosa cells of secondary follicles (SF) were analyzed ($p < 0.05$).

12) Fig. 3E should be included as a Supplemental table instead.

According to your suggestion, we moved Figure 3E to Supplemental Table 1,2.

13) In S2a-c showing OCR, H3K27ac and transcript expression changes, the graphs of genes should be put in the same order as shown in S1-B.

In revised manuscript, the list of the top 9 most demethylated genes was moved to supplemental Table 1. In supplemental Figure 3 (a)-(c), the graph of genes was put in the same order as the Supplemental Table 1 according to your suggestions.

14) Line 215 and Fig. S3-c. It is very abrupt and unclear as to why the AREG experiment was included to look at H3K27ac of LH target genes. If not necessary, this information needs to be omitted from the paper for simplification of results and interpretation.

According to your suggestions, we deleted the data of the AREG study from Supplemental Figure 3. The sentences in result section that mention these data were also deleted.

15) Lines 226 to 237. The word alleles need to be replaced by CpG sites. Line 235-237, the authors are making a presumptive statement by directly attributing the demethylation of FSH target genes to DNMT1 downregulation and TET2 upregulation. This sentence needs to be expressed differently eg: by using the words may be or possibly.

According to your suggestion, we replaced the word alleles to CpG sites. Additionally, we added “possibly” to the sentence as follows.

Page 7, line 224-226

These results indicated that demethylation was linked to cell cycle progression and was possibly related to changes in the cellular levels and activity of DNMT1 and TET2 (Figure 1).

16) Line 392 – TETS do not inactivate the methyl group. It participates in demethylation of CpG and non-CpG bases via oxidative demethylation mechanisms. The sentence should be rewritten.

Acceding to your suggestion, we rewrote this sentence as follows.

Page 9, line 276-278

When TET factors are recruited to methylated regions of the DNA during S phase, TET converts 5mC to 5hmC via oxidative demethylation mechanisms leading to demethylation of methylated DNA regions [41].

17) Lines 341-353 do not fit in the context of the paper and are just adding to the bulk in discussion. These can be removed or simplified.

Acceding to your suggestion, we have removed most of sentences and then focused on the discussion about the epigenetic regulation of LH-target genes as follows.

Page 10, line 313-Page 11, line 331

Interestingly, in the genes that were induced in preovulatory granulosa cells in response to hCG stimulation of ovulation, the demethylation of promoter regions had already occurred. In the promoters of the LH target genes, Star and Cyp11a1, Histone-H4 acetylation (Ac-H4) and trimethylation of histone-H3 lysine-4 (H3K4me3) are increased, whereas H3K9me3 and H3K27me3 are decreased in luteinized granulosa cells after hCG injection [47]. Of critical importance, granulosa cells isolated from small follicles are not capable of undergoing luteinization when cultured with LH or forskolin+PMA, compounds that mimic LH signaling in preovulatory granulosa cells [48]. By contrast, granulosa cells isolated from preovulatory follicles not only express high levels of the LH receptor but also rapidly differentiate into luteinized cells by both treatments [48], suggesting that the histone modifications are induced by LH-stimulating signaling pathways in promoter regions where DNA demethylation and changes of DNA structure had already occurred. The histone modifications and the expression of LH target genes critical for ovulation were only induced by hCG in this study as shown in previous reports [49]. By contrast, in FSH-target genes that were induced in granulosa cells of growing follicles by eCG, DNA demethylation, changes of DNA structure and histone modifications occurred at the same time points. The timing of histone modification is

dependent on the types of transcription factors that are activated by different signaling pathways and recruited to histone acetylation factors [50,51]. Thus, the order of DNA demethylation, changes of DNA structure and then histone acetylation appears to be strictly determined in each promoter region in granulosa cells.

18) Lines 354-355 – the sentence that the promoters of the three LH target genes genes are tightly packed needs to be rewritten. Consider writing hypermethylated instead. But overall the paragraph from lines 354-368 are important findings of the paper and are highly appreciated.

According to your suggestion, we changed the word “tightly packed” to the word “hypermethylated”. The paragraph was restructured to bind the previous paragraph and emphasize its importance as shown in your above suggestion.

19) Lines 369-371-The statement that large scale demethylation did not occur in cumulus cells is not seen in the data provided. This sentence needs to be removed as it borders on over-interpretation of data.

According to your suggestion, we deleted this sentence.

20) Fig 6 and Fig 7 and the related results and discussion is very very complicated to go through because of the a,b style of significance shown in bar graphs and the complicated design of experiments. The reader loses focus of the manuscript here.

21) Overall, the paper needs to be simplified beyond the point where Fig 6 begins.

To simplify Figure 6 and Figure 7 and make them easier to understand, we excluded the data of COCs cultured with the SMAD inhibitor. The treatment groups of cultured granulosa cells were arranged to make it easier to understand the role of RA. In order to show the effects of RA and/or FSH + T, the significant differences compared to the control (without any factors) are shown as * ($p < 0.05$) in Figure 7 A.

Reviewer #3

1. Introduction is too long, and the major hypothesis is lost.

Version correctly in response to previous reviewers.

According to your and other reviewer's suggestion, we have revised the introduction to state the aim of this study, first, the hypothesis that a large-scale change in the DNA methylation status would be induced by both a decrease in the amount of DNMT1 and cell division occurring at the same time, which would induce cell differentiation was mentioned. Second, we also mentioned that during ovarian follicle development, granulosa cells undergo functional changes after cell division, and that demethylation of the promoter regions occurs in a retinoic acid-dependent manner in *Lhcgr* expression, that is a marker of granulosa cell differentiation. Therefore, we stated that the aims of this study were to determine the relationships among cell division, retinoic acid, DNMT1 expression and activity and epigenetic regulation of gene structure using granulosa cells as a model.

Please read our new introduction section (Page 2, line 40-Page 4, line 97).

2. Methods part is not clear, the authors did not provide sufficient details of the experiments, so it is hard to interpret some data.

According to your suggestions, we added an "experimental design" section to the Materials and Methods section. Detailed information about sample collection was also included in new Materials and Methods section and with schematic in Supplemental Figure 6. The experimental method was also explained in detail, including the following points.

3. In animal study, the treatment doses, the number of animals for each treatment group, the drug dissolving and delivery, the manufacturer of the drug were not described.

According to your suggestion, we have added the information requested.

Page 11, line 373-age 12, 385

Immature female (3-week-old) C57BL/6 mice were obtained from Charles River Laboratories Japan (Yokohama, Japan). Twenty-two-day-old female mice were injected intraperitoneally with 4 IU of eCG to stimulate follicular growth; after 48 hours, they were injected with 5 IU of hCG to stimulate ovulation and luteinization. In our experiments, granulosa cells or ovaries were collected from 2 to 3 mice in each treatment group for real-time PCR analysis, western blotting, immunofluorescence staining. For flow cytometry, granulosa cells were collected from two mice in each treatment group, and four mice were used for the bisulfite sequence assay, MeDIP-Seq analysis, ChIP assay and

FAIRE-qPCR analysis in each treatment group. All of studies were repeated at least 3 times. The animals were housed under a 12-h light/12-h dark schedule in the Experimental Animal Center at Hiroshima University and provided with food and water ad libitum. The animal study was approved by the Hiroshima University Animal Committee (Permit Number: C18-34), and the mice were maintained in accordance with the Hiroshima University Guidelines for the Care and Use of Laboratory Animals.

Page 14, line 461-468

In vitro culture of granulosa cells

Granulosa cells were collected from ovaries of immature (3-week-old) mice at 6 h following eCG injections [25]. Cells were seeded onto an FBS-recoated 24 or 96 well plates. Granulosa cells were treated with 100 ng/ml of FSH (NIDDK, Torrance, CA)/DMEM/F12 medium, testosterone (10 ng/ml, Sigma)/ethanol, and/or retinoic acid (1 μ M, Sigma)/DMSO in the presence or absence of 1 % (v/v) FBS. Some granulosa cells were treated with 0.05, 0.5, 5 μ M aphidicolin (Sigma)/DMSO in the presence of FSH, testosterone and 1 % (v/v) FBS. Other granulosa cells were co-cultured with denuded oocytes according to our previous study [25].

4. In RNA extraction method, it is not clear how much of total RNA was used for RT QPCR, did authors treat RNA with DNase?; the use Qiagen RNeasy mini kit, does not guarantee the absence of DNA.

We treated RNA with DNase (RNase-Free DNase Set, #79254, QIAGEN). 50 ng/ μ l of total RNA were used for cDNA conversion and 3 μ l of cDNA was used for real-time PCR analysis. This information was described in materials and methods as follows.

Page 12, line 394-402

RNA extraction and real-time PCR

Total RNA was obtained from mouse granulosa cells of secondary follicles with multilayered granulosa cells (Type 5b), mouse granulosa cells or COCs (from antral or PO follicles ???) using RNAeasy Mini Kit (Qiagen Sciences, Germantown, MD, USA) according to the manufacturer's instruction and our previous study [56]. The RNA samples were treated with DNase (Qiagen), and 50 ng/ μ l of total RNA were reverse transcribed in 20 μ l reaction buffer. Three μ l of cDNA were used for real-time PCR study. The primer sets are shown in Supplemental Table 5. L19 was used as a control for reaction efficiency and variations in concentrations of mRNA in the original RT reaction.

5. MeDIP-seq, how DNA was sonicated, instrument, parameters? What was the average size of the

fragment after sonication, which antibody (reference and manufacturer) was used? How many biological replicates were used for each group? what is cut-off parameter for differential methylation analysis? Whether the multiple test (FDR) was applied, if yes, which cut -off p-value was used? If there was any internal quality control, such as analysis of methylation of imprinted genes or spike-in control used?

For MeDIP-seq analysis, DNA samples were sonicated by Covaris S2 (Covaris) under following conditions (Setting: Value, Duty Cycle:10 %, Intensity:5, Cycles per Burst: 200, Time: 2 cycles of 60 seconds each). The average size of the fragment after sonication was 350 bp. The fragmented DNA was dissociated into single strands, then the methylated region was enriched by Methyl-CpG binding domain of human MBD2 protein using MethylMiner Methylated DNA Enrichment Kit (Invitrogen, [59, 60]). Quantification of DNA methylation at multiple FDR adjusted p-value ($p < 0.05$) cut-off. For MeDIP-seq analyses, we analyzed 3 biological replicates for each group. The analysis of methylation of imprinted genes was shown in Supplemental Table 2. The results showed that methylation status of imprinted genes was not significantly changed in granulosa cells during follicular development. This information was added to Materials and Methods as follows:

Page 12, line 410-Page 13, line 427

One μg of DNA was used as an input sample and treated with MethylMiner Methylated DNA Enrichment kit (Invitrogen). Three biological replicates were used for each group. The DNA was sonicated by Covaris S2 (Covaris, Woburn, MA) under following conditions (Setting: Value, Duty Cycle:10 %, Intensity:5, Cycles per Burst: 200, Time: 2 cycles of 60 seconds each). The average size of the fragment after sonication was 350 bp. The fragmented DNA was dissociated into single strands, then the methylated region was enriched by Methyl-CpG binding domain of human MBD2 protein using MethylMiner Methylated DNA Enrichment Kit (Invitrogen, [59, 60]). Quantification of DNA methylation at multiple FDR was adjusted to p-value ($p < 0.05$) cut-off. TruSeq DNA HT Sample Prep kit (Illumina, Tokyo, Japan) was used for the Paired End library method. The sequence of the collected sample was decoded using an Illumina HiSeq 2000, and the methylated region was identified by MEDIPS software. Absolute methylation signals (AMS) were used for judging DNA methylation levels of the promoter region in each gene. The prediction of gene function was analyzed using Functional Annotation Bioinformatics Microarray Analysis (<https://david-d.ncifcrf.gov/tools.jsp>). The accession number for all sequence data shown in this paper is DNA Data Bank of Japan (DDBJ) Sequence Read Archive: DRA010809 (http://trace.ddbj.nig.ac.jp/dra/index_e.html). The analysis of methylation of imprinted genes was showed in Supplemental Table 2 as a control.

6. ChIP assay, how many biological replicates were used for each group, what is the size of the sheared chromatin, which negative and positive control for ChIP were used? The quantity of antibody used for ChIP and the reference and manufacturer of the antibody must be provided.

For Chip assay, we analyzed with 3-4 biological replicates. The size of the sheared chromatin was 100bp-1000bp. Normal rabbit IgG antibody (Cell signaling 2729) as negative control and Histone H3 (D2B12) XP® Rabbit mAb (ChIP Formulated) (Cell signaling 4620) as positive control were used in this study. These antibodies were included in SimpleChIP® Enzymatic Chromatin IP Kit (Magnetic Beads) (Cell signaling#9003). These antibodies were used at the same dilution as H3K27ac antibody (1:100). The conditions used for the Chip assay were described in Materials and Methods as follows (Page 13, line425-436). The quantity of antibody used for ChIP, the reference and manufacturer of the antibody were shown in Supplemental Table 7.

Page 13, line 441-452

Chip assay

The DNA–protein complexes were collected from secondary follicles with multilayered granulosa cells (Type 5b) or granulosa cells from mouse (3-week-old) ovaries at 0, 24, 48 h after eCG injection or at 4, 8 h after hCG injection. Chip assay was done using SimpleChIP® Enzymatic Chromatin IP Kit (Magnetic Beads) (Cell signaling, MA, USA)) according to the manufacturer’s instructions. Three or four biological replicates were performed for each group. One hundred bp-1000 bp of the sheared chromatin was obtained by Ultrasonic disruptor (TOMY UD-200). ChIP was performed using anti-Histone H3 containing the acetylated lysine 27 (H3K27ac) antibody (1:100) (Cell signaling). Normal rabbit IgG antibody (1:100) (Cell signaling) as negative control and Histone H3 (D2B12) XP® Rabbit mAb (ChIP Formulated) (1:100) (Cell signaling) as positive control was used. The primer sets used for the detection of each specific region are shown in Supplemental Table 7.

7. Figure1, WB analysis of H3K27ac is not convincing. For histone detection by WB, the histone purification prior of WB is recommended.

Thank you for your suggestion. By extending the sonication time, we detected a sharp band and with this new condition, we repeated the experiment three times. These results are shown in Supplemental Figure 4D.

8. Fig 1C, Immunofluorescence pictures, the manufacturers of the antibodies and the references must

be provided. Did authors use the same exposure time for image acquisition? How many images did the authors analyze and how many replicates?

Thank you for your suggestion. We showed the information of the manufactures of the antibodies and the references in Supplemental Table 7. Primary antibodies (1:100 anti-PCNA antibody, Cell signaling#2586S, 1:100 anti-DNMT1 antibody, GeneTex#GTX116011, 1:100 anti-TET2 antibody, Abcam#ab124297) were used. We used the same exposure time for image acquisition. For immunofluorescence study, we used 1 mouse in each experiment. Immunostaining of DNMT1 and PCNA, TET2 and PCNA was performed using continuous sections. At least 3 follicles in each ovary were used for counting the number of positive cells. In each treatment group, three mice were used as triplicated experiments.

9. It is difficult to appreciate the Figure 3E as there is no connection between the table of functional annotation and the observed effects that were discussed. The authors should discuss the detected alterations in group of genes combined by common function, eg. “Chromatin modification” and the possible impact on biological effects which were observed.

Thank you for your suggestion. We added the discussion as follows.

Page 9, line 293-Page 10, line 312

To identify the promoters of genes that were selectively demethylated in S-phase during granulosa cell differentiation, MeDIP-Seq analyses were done. Highly demethylated promoters accounted for about 40 % of the whole genome including not only well-known granulosa cell markers but also promoters of other genes. Annotation analysis predicted changes in metabolic processes, phosphorylation/signaling pathways, cytoskeletal organization, transmembrane transport, catabolic processes and chromatin modification. Factors involved in transcription and DNA modification were also significantly changed. During follicular development, mitochondrial ATP production that is required for cell proliferation is dominant and increased in granulosa cells by eCG [42]. Multiple signaling pathways are known to be activated in granulosa cells during follicular development, including serine/threonine kinase and/or PI3K/AKT pathways [43]. In the top 9 demethylated genes, STK36 (serine/threonine kinase 36) inhibits the activation of Gli factors that are activated by the hedgehog signal transduction cascade and regulate cell proliferation and tumorigenesis [44]. Tranu1ap (tRNA selenocysteine 1 associated protein 1) has been shown to inhibit proliferation in cancer cells by acting through the PI3K/AKT pathway [45]. Moreover, it is also well known that in granulosa cells, changes in the cytoskeleton and changes in cell membrane transport occur during follicular development [46], suggesting that dynamic changes in DNA demethylation within specific

promoter regions of the whole genome are essential for granulosa cell proliferation and differentiation. However, the functions of most highly demethylated genes detected in this study including the top 9 genes, such as Stk36 and Trna1ap remain to be determined.

10. Discussion is very long, and it is often the repetition of the Results parts. It should be reduced to a concise version.

Thank you for your suggestion. We reconstructed the Discussion part, specially deleted an expanded repetition of the results.

Reviewers' comments:

Reviewer #1 (Remarks to the Author):

The writing of this manuscript needs to be strengthened, and there are still many problems to be explained:

- 1.The paper did not examine the relationship between DNA demethylation and granulosa cell differentiation, so the title is still inaccurate.
- 2.The second paragraph in the introduction (Line 52-66) is not clear and not relevant to the content of the manuscript, so it should be removed. In addition, references 9-13 do not enable the reader to draw scientific hypotheses similar to those of the author (Line 61-63).
- 3.Line 76-79, *Lhcgr* is not only expressed in FSH-stimulated granulosa cells, but is induced during the development of preantral follicles.
- 4.Line 88-97, the logic of the paragraph is so confusing that it is difficult to read.
- 5.Line 173, what is the purpose of the authors' deliberate emphasis that *Amhr2*, *Gja1*, *Nr5a1*, and *Lhcgr* are eCG-induced genes? Are *Fshr* and *Ccnd2* not induced by eCG (Line 170).
- 6.Line 195-209, the authors claim that demethylation and altered chromatin structure induced by eCG were required for subsequent histone modifications and the induction of LH-target genes. However, the experimental data are not sufficient to support this view, the authors did not test whether hCG can independently induce histone modification without eCG pretreatment.
- 7.Line 197, in fact, *Star*, *Ptgs2* and *Cyp11a1* are both target genes of LH and FSH. Why did the author not select specific target genes of LH to study?
- 8.Fig. 6, I still insist on suggesting the author to clarify which oocyte factor (GDF9 or BMP15) affects the expression of *Dnmt1* and *Tet2* in granulosa cells. Technically, the problem is easy to solve ;
- 9.In Fig.6, FSH+T significantly inhibited the expression of *Dnmt1*, while in Fig.7, FSH+T did not affect the expression of *Dnmt1*. What is the cause of this contradiction.
- 10.In Fig. 7A, RA was sufficient to independently induce the expression of *Dnmt1* and *Tet2*. So is the demethylation of granulosa cells during gonadotropin-dependent folliculogenesis induced by RA? What is the relationship between eCG and RA? Is RA a target molecule regulated by eCG?
- 11.Thanks to the authors for showing us that eCG stimulates the demethylation of granulosa cells, but what is the physiological significance of this demethylation? In other words, what does it mean for granulosa cell proliferation and differentiation? What does it mean for folliculogenesis and ovulation? These are the questions readers want to know more about.

Reviewer #2 (Remarks to the Author):

The authors have provided reasonable answers to our previous queries and have addressed the changes satisfactorily. The study is highly appreciated as it provided new insights on the epigenetic modifications occurring in important ovarian genes during different stages of follicular growth and/or folliculogenesis. The data showing robust structural changes in chromatin configuration, DNA methylation levels and H3K27 acetylation status of *Stk36*, *Lhcgr* and *Trna1ap* are important to the field of endocrinology. We therefore recommend the article for consideration towards publication provided that the following changes are made to the manuscript.

- Lines 52-60. Here, the context of introduction suddenly shifts to epithelial-mesenchymal transition reported in different cell types or conditions. We understand that the authors are referring to EMT since this is one of the proposed mechanisms by which the granulosa cells develop during follicle growth. However, this is a widely debated topic with several theories under consideration. Therefore, it is recommended that the authors either provide a background on why it is EMT is important in the context of granulosa cells or simply omit this part from the introduction.
- For lines 60-61...widespread DNA demethylation is associated with the reduction of DNMT1 expression and/or the decrease of DNMT1 enzyme activity, the reference/s have not been cited.
- Line 109, instead of "other factors", please state the names of genes i.e. *Dnmt3a*, *Uhrf1*, *Tet1* and *Tet3* whose expression levels remained unchanged.
- Line 399: Instead of the concentration of total RNA, the authors may consider stating the

absolute amount of RNA that was used for conversion to cDNA eg. 1 μ g.

- For figure 4, indicate (L19) at the Y axis (as given for Fig.1) where the graphs for qPCR are provided. It is recommended that the authors maintain consistency throughout the figures and/or tables within the manuscript, wherever applicable.
- For figure 5, the lollipop maps for CpG islands with hypo or hypermethylated CpG sites can be combined together with a partition for each of the three genes for easier visual comparison. i.e eg: for Lhcgr SF, eCG-0hr, 24hr, 48hr (partition) C, FSH+T, FSH+T+aphi. The tables can be similarly arranged side by side with a partition.
- Supplemental figure 4 legend, instead of stating the term kinetic changes seems misleading, and should be replaced with the term "temporal changes" instead.
- The document needs to be rechecked for grammar and repetitions of certain words eg: lines 86-87 (during the is repeated)
- Caveats of the study and future plans should be discussed at the end of the discussion section.

Reviewer #3 (Remarks to the Author):

The authors addressed all raised points and substantially revised the manuscript and that makes the story clearer and more logical. I suggest to accept it.

Responses to reviewer' s comments

Reviewer #1 (Remarks to the Author):

1. The paper did not examine the relationship between DNA demethylation and granulosa cell differentiation, so the title is still inaccurate.

According to your suggestion, the title is changed to “Large-scale DNA demethylation occurs in proliferating ovarian granulosa cells during follicular development”

2. The second paragraph in the introduction (Line 52-66) is not clear and not relevant to the content of the manuscript, so it should be removed. In addition, references 9-13 do not enable the reader to draw scientific hypotheses similar to those of the author (Line 61-63).

The second paragraph in the introduction was removed. The hypotheses in this paper are summarized as follows;

1. Cell proliferation might be involved in the differentiation of granulosa cells and coupled to the largescale epigenetic changes that occur during follicular development.
2. DNA methylation patterns might not be completely copied during cell proliferation and specific changes in DNA methylation patterns would induced in granulosa cells during follicular development by specific regulatory mechanisms.

Please read the revised introduction.

3. Line 76-79, *Lhcgr* is not only expressed in FSH-stimulated granulosa cells, but is induced during the development of preantral follicles.

Lhcgr is induced in theca cells that are components of the follicular wall from the secondary follicle stage to the preovulatory stage. In our previous study (Kawai et al., 2018, Endocrinology), the methylation status of the *Lhcgr* promoter region in theca cells was lower than that in granulosa cells in small antral follicles and the level in theca cells did not change during follicular development. This information was added in introduction section as follows.

Page 3, line 72-76; In contrast, *Lhcgr* is constitutively expressed in theca cells of growing and

preovulatory follicles, and the DNA methylation rate is low [19], indicating that the DNA methylation pattern might be altered in granulosa cells but not in theca cells during the marked transition of granulosa functions in preovulatory follicles.

4. Line 88–97, the logic of the paragraph is so confusing that it is difficult to read.

The final paragraph has been revised to simple sentences that introduces the aims of the research as follows.

Page 3, line 80–85; The reduction of DNA methylation in the *Lhcgr* promoter region of granulosa cells is dependent on not only FSH but also the de novo synthesis of retinoic acid (RA) and SMAD pathways [19]. RA and SMADs are known factors involved in cell fate determination due to their impact on epigenetic modifications [21,22]. Therefore, the studies described herein were undertaken to analyze granulosa cells as a model to determine the underlying mechanisms by which DNA methylation changes dramatically in highly proliferative cells.

5. Line 173, what is the purpose of the authors' deliberate emphasis that *Amhr2*, *Gjal*, *Nr5a1*, and *Lhcgr* are eCG-induced genes? Are *Fshr* and *Ccnd2* not induced by eCG (Line 170).

The functions of many of the genes that change significantly in granulosa cells during follicular development have not been reported. Therefore, we selected genes whose expression and function in granulosa cells during follicular development are known. In the genes that have high levels of expression maintained in granulosa cells, the methylation status did not change significantly; however, in the genes where the methylation status decreased, the expression levels were significantly increased during follicular development. To provide background for the readers to understand that the large-scale changes in methylation occur in granulosa cells and play an important role in follicular development, we provided data describing that the reduction of DNA methylation status occurred in the genes in which transcription is activated in granulosa cells.

6. Line 195–209, the authors claim that demethylation and altered chromatin structure induced by eCG were required for subsequent histone modifications and the induction

of LH-target genes. However, the experimental data are not sufficient to support this view, the authors did not test whether hCG can independently induce histone modification without eCG pretreatment.

LH receptor levels in granulosa cells of preantral and small antral follicles are negligible and therefore, LH does not impact the functions of granulosa cells in these follicles before eCG treatment. Thus, it is difficult to judge whether hCG can directly induce the histone modification or not. The title in this section was changed as follows.

Page 6, line 180-181; Histone modification and enhanced gene expression occurred only after hCG injection following eCG-induced demethylation and altered chromatin structure in LH target genes

In the final sentence of this paragraph, the conclusion was revised to as follows:

Page 6, line 190-192; Increased acetylation occurred simultaneously with significantly increased expression of the LH target genes in eCG-primed granulosa cells exposed to an ovulatory dose of hCG (Figure S3C).

7. Line 197, in fact, *Star*, *Ptgs2* and *Cyp11a1* are both target genes of LH and FSH. Why did the author not select specific target genes of LH to study?

These genes are expressed at very low levels in granulosa cells in preovulatory follicles; however, their induction is dramatically increased in periovulatory follicles after hCG injection and is essential for ovulation and the formation of functional corpora lutea that produce progesterone. Neither LH nor FSH can induce their expression to the same extent in granulosa cells from small antral follicles. Before the ovulatory induction of these genes (*Star*, *Ptgs2* and *Cyp11a1*) in response to hCG injection, mRNA encoding the EGF-like factors was dramatically induced over 100 fold in granulosa cells and participated in mediating the hCG mediated transcriptional events in preovulatory follicles (Shimada et al., Mol Endocrinol, 2006). Thus, we did the additional study to analyze the epigenetic changes of *Areg*, *Ereg* and *Btc* in granulosa cells before or after hCG injection and the data were shown in Figure S3.

8. Fig. 6, I still insist on suggesting the author to clarify which oocyte factor (GDF9 or BMP15) affects the expression of *Dnmt1* and *Tet2* in granulosa cells. Technically, the problem is easy to solve ;

According to your suggestions, we cultured granulosa cells with GDF9. The data clearly showed that GDF9 strongly suppressed FSH-induced the reduction of *Dnmt1* expression and FSH-induced the expression of *Tet2* in granulosa cells. The data were shown in Figure 6B.

Page 7, line 218-221; The reduced expression of *Dnmt1* and the induction of *Tet2* were also regulated by additional treatment with GDF9 (Figure 6B). However, the addition of RA to the medium overcame the negative effects mediated by coculture with denuded oocytes or treatment with GDF9 (Figure 6A, B).

9. In Fig.6, FSH+T significantly inhibited the expression of *Dnmt1*, while in Fig.7, FSH+T did not affect the expression of *Dnmt1*. What is the cause of this contradiction.

In figure 6, granulosa cells were cultured in the presence of serum. FSH+T induced the expression of ADH and ALDH that are key enzymes for the production of RA from serum-containing retinol. In Figure 7, the granulosa cells were cultured in the absence of serum. Therefore, the addition of RA was required for the reduction of *Dnmt1* expression. In Figure 7B, the reduction of DNMT1 at S phase was suppressed by 4MP, an inhibitor for ADH, when granulosa cells were cultured with FSH+T under the presence of serum. The data support our conclusion.

10. In Fig. 7A, RA was sufficient to independently induce the expression of *Dnmt1* and *Tet2*. So is the demethylation of granulosa cells during gonadotropin-dependent folliculogenesis induced by RA? What is the relationship between eCG and RA? Is RA a target molecule regulated by eCG?

In our previous study (Kawai et al., 2016, Endocrinology), it was reported that RA was produced from retinol in granulosa cells. The production was dependent on the functions of two enzymes, ADH and ALDH that were expressed in granulosa cells by FSH stimulation. Thus, as you think, RA is a factor produced by eCG stimulation and is a factor that plays a part in the action of eCG. In our previous study we also reported

that the expressions of ADH and ALDH were significantly suppressed by co-culture with denuded oocytes. Therefore, to explain the relationship between RA and eCG stimulation, we added the sentences in discussion section as follows.

Page 10, line 324-331; These effects on oocytes were also induced by treatment with GDF9, and their negative effects were completely overcome by exogenous RA. RA is produced from retinol in 2 steps [55], and the reactions are dependent on the expression of ADH and ALDH in FSH-stimulated granulosa cells during follicular development [56]. Importantly, the induction of both enzymes is suppressed in granulosa cells by coculture with oocytes [19]. Thus, the proliferation of granulosa cells that leads to an increase in follicle diameter might be required to allow RA synthesis by reducing the local concentration of oocyte-secreted factors in follicular fluid.

11. Thanks to the authors for showing us that eCG stimulates the demethylation of granulosa cells, but what is the physiological significance of this demethylation? In other words, what does it mean for granulosa cell proliferation and differentiation? What does it mean for folliculogenesis and ovulation? These are the questions readers want to know more about.

We added the sentences to introduce our idea of what role DNA demethylation plays in folliculogenesis and ovulation as follows.

Page 10, line 332-343; In antral follicles, follicular fluid accumulates within the follicle that separates granulosa cells from the enclosed cumulus cell-oocyte complex. Because cumulus cells are strongly regulated by the oocyte via oocyte-secreted factors that activate SMAD pathways [57], RA production is limited [19], and a high level of Dnmt1 was observed in cumulus cells of preovulatory follicles. Thus, the distance from the oocyte determines the epigenetic status in follicular somatic cells and their fate as cumulus cells or granulosa cells. In other words, cell proliferation first indirectly weakens the mechanisms of precise copying of the DNA methylation status due to the distance from oocyte and second, cell proliferation directly changes the methylation status and transcriptome in granulosa cells during the follicular development process. Collectively, these results indicate that the epigenetic regulation of granulosa cell differentiation mediated by cell proliferation, RA and oocyte-secreted factors is one of the most highly orchestrated processes in female reproduction.

Reviewer #2 (Remarks to the Author):

· Lines 52-60. Here, the context of introduction suddenly shifts to epithelial-mesenchymal transition reported in different cell types or conditions. We understand that the authors are referring to EMT since this is one of the proposed mechanisms by which the granulosa cells develop during follicle growth. However, this is a widely debated topic with several theories under consideration. Therefore, it is recommended that the authors either provide a background on why it is EMT is important in the context of granulosa cells or simply omit this part from the introduction.

According to your and other reviewer's suggestions, we removed the paragraph about EMT in introduction section.

· For lines 60-61...widespread DNA demethylation is associated with the reduction of DNMT1 expression and/or the decrease of DNMT1 enzyme activity, the reference/s have not been cited.

The sentence was removed.

· Line 109, instead of "other factors", please state the names of genes i.e. Dnmt3a, Uhrf1, Tet1 and Tet3 whose expression levels remained unchanged.

According to your suggestion, the names of genes were mentioned in this sentence as follows.

Page 4, line, 98-99; The expression levels of Dnmt3a, Uhrf1, Tet1 and Tet3 did not change significantly during preovulatory follicle growth (Figure 1A).

· Line 399: Instead of the concentration of total RNA, the authors may consider stating the absolute amount of RNA that was used for conversion to cDNA eg. 1 • g.

We mentioned the amount of RNA for conversion to cDNA as follows.

Page 12, line 410-411; 50 ng of total RNA was reverse transcribed in 20 ul reaction buffer.

- For figure 4, indicate (L19) at the Y axis (as given for Fig.1) where the graphs for qPCR are provided. It is recommended that the authors maintain consistency throughout the figures and/or tables within the manuscript, wherever applicable.

According to your suggestions, All gene expression data (qPCR) have been unified to /19 at Y axis.

- For figure 5, the lollipop maps for CpG islands with hypo or hypermethylated CpG sites can be combined together with a partition for each of the three genes for easier visual comparison. i.e eg: for Lhcgr SF, eCG-0hr, 24hr, 48hr (partition) C, FSH+T, FSH+T+aphi. The tables can be similarly arranged side by side with a partition.

According to your suggestions, we corrected the order of figures.

- Supplemental figure 4 legend, instead of stating the term kinetic changes seems misleading, and should be replaced with the term “temporal changes” instead.

We changed the term “kinetic” to “temporal” in this figure legend.

- The document needs to be rechecked for grammar and repetitions of certain words eg: lines 86-87 (during the is repeated)

We rechecked the grammar errors in our manuscript.

- Caveats of the study and future plans should be discussed at the end of the discussion section.

We added the consideration of research possibilities as follows.

Page 10, line 344 to page 11, line 361;

The maturation and developmental competence of oocytes decrease with increasing age in not only female mice but also women [58,59]. One of the reasons has been reported to be that the level of oocyte-secreted factors is lower in oocytes recovered from aged mice than in oocytes recovered from younger mice [60]. In older infertility patients, abnormal luteinization and low quality of oocytes in small antral follicles have been observed [61,62]. The decreasing ovarian functions in older women/female mice would be involved in abnormal promoter DNA

methylation of critical genes in granulosa cells and cumulus cells, based on our evidence that oocyte-secreted factors strongly regulate the epigenetic changes in both cells. Moreover, bacterial infections of the female genital tract result in pelvic inflammatory disease (PID), which causes infertility [63]. The injection of lipopolysaccharide (LPS), which is a component of gram-negative bacteria, alters the DNA methylation status in the Lhcgr promoter region in granulosa cells of the mouse ovary [64]. Based on the present study that focused on the changes in epigenetic status in granulosa cells during follicular development in immature mice, it is expected that analyses of the epigenetic status in granulosa cells will make it possible to identify potential causes of reproductive disorders and infertility. In particular, the analysis of DNA methylation in granulosa cells may lead to the development of new treatments and preventions for infertility or the development of contraceptives because DNA methylation status is an index (or measure) of whether gene expression is possible or not.

REVIEWERS' COMMENTS:

Reviewer #1 (Remarks to the Author):

I appreciate that the authors have addressed all my previous concerns,I have no further concerns.